# KChIP2 is a core transcriptional regulator of cardiac excitability

Drew M Nassal[1,2], Xiaoping Wan[1], Haiyan Liu[1], Danielle Maleski[1], Angelina Ramirez-Navarro[1], Christine S Moravec[3], Eckhard Ficker[1†], Kenneth R Laurita[1], Isabelle Deschênes[1,2]*

[1]Heart and Vascular Research Center, Department of Medicine, Case Western Reserve University, Cleveland, United States; [2]Department of Physiology and Biophysics, Case Western Reserve University, Cleveland, United States; [3]Department of Molecular Cardiology, Cleveland Clinic, Cleveland, United States

**Abstract** Arrhythmogenesis from aberrant electrical remodeling is a primary cause of death among patients with heart disease. Amongst a multitude of remodeling events, reduced expression of the ion channel subunit KChIP2 is consistently observed in numerous cardiac pathologies. However, it remains unknown if KChIP2 loss is merely a symptom or involved in disease development. Using rat and human derived cardiomyocytes, we identify a previously unobserved transcriptional capacity for cardiac KChIP2 critical in maintaining electrical stability. Through interaction with genetic elements, KChIP2 transcriptionally repressed the miRNAs miR-34b and miR-34c, which subsequently targeted key depolarizing ($I_{Na}$) and repolarizing ($I_{to}$) currents altered in cardiac disease. Genetically maintaining KChIP2 expression or inhibiting miR-34 under pathologic conditions restored channel function and moreover, prevented the incidence of reentrant arrhythmias. This identifies the KChIP2/miR-34 axis as a central regulator in developing electrical dysfunction and reveals miR-34 as a therapeutic target for treating arrhythmogenesis in heart disease.

*For correspondence: isabelle.
deschenes@case.edu

†Deceased

**Competing interests:** The authors declare that no competing interests exist.

## Introduction

Cardiac excitability is controlled by a combination of depolarizing and repolarizing currents, whose dysregulation during heart failure (HF) or myocardial infarction (MI) play a significant role in clinically relevant arrhythmias (*Tomaselli and Marbán, 1999*; *Wang and Hill, 2010*). Aberrant remodeling culminates in altered $Ca^{2+}$ current ($I_{Ca}$) (*Houser et al., 2000*; *Wang et al., 2008*), $Na^+$ current ($I_{Na}$) (*Pu and Boyden, 1997*; *Maltsev et al., 2002*), and a host of outward $K^+$ currents ($I_K$) (*Näbauer and Kääb, 1998*), creating impaired cardiac excitability and performance, accounting for high rates of mortality in HF patients (*Nattel et al., 2007*; *Tomaselli and Zipes, 2004*). However, large variability and breadth of electrical changes present challenges in determining which mechanisms are critical in driving arrhythmias and disease progression. Intriguingly, loss of the Potassium Channel Interacting Protein 2 (KChIP2) has proven to be a consistent event following cardiac stress, sparking interest into understanding its contribution in disease remodeling (*Näbauer and Kääb, 1998*; *Nass et al., 2008*; *Jin et al., 2010*).

It is well described that KChIP2 associates with and modulates the Kv4 family of potassium channels, which together define the fast transient outward potassium current ($I_{to,f}$), maintaining early cardiac repolarization (*An et al., 2000*; *Niwa and Nerbonne, 2010*). However, emerging evidence suggests KChIP2 may not be limited to this role (*Thomsen et al., 2009*; *Li et al., 2005*; *Deschênes et al., 2008*). Investigations following KChIP2 knock-down show reduced transcript expression for the cardiac sodium channel gene, *SCN5A*, and its accessory subunit *SCN1B*, in

**eLife digest** The heart pumps blood throughout the body to provide oxygen and nourishment. To do so, proteins in the heart create electrical signals that tell the heart muscles to contract in a coordinated manner. Heart disease can cause cells to lose control of the production or activity of these proteins, creating disorganized electrical signals called arrhythmias that interfere with the heart's ability to pump. Sometimes these arrhythmias lead to sudden death.

Researchers do not know exactly what triggers these changes in the heart's normal electrical rhythms. This has made it difficult to develop strategies to prevent these disruptions or to fix them when they occur.

By studying rat and human heart cells, Nassal et al. now show that a protein called KChIP2 stops working properly during heart disease. Most importantly, because of the decreased level of KChIP2 in heart disease, KChIP2 loses the ability to restrict the production of two microRNA molecules – a role that KChIP2 was not previously known to perform. This loss of activity sets off a cascade of signals that worsens the balance of electrical activity in the heart cells, creating arrhythmias.

Treatments that restored proper levels of the fully working KChIP2 protein to the heart cells or that blocked the signals set off by a lack of KChIP2 returned the electrical activity of the cells back to normal. This also stopped the development of arrhythmias. Further studies are now needed to investigate whether these treatments have the same effects in living mammals. If effective, this could ultimately lead to new treatments for heart diseases and arrhythmias.

addition to Kv4.3 protein, prompting the loss of both $I_{to,f}$ and $I_{Na}$ (*Deschênes et al., 2008*). Considerably, these changes reflect conditions observed in the diseased heart, but more importantly implicate potential transcriptional significance for KChIP2 at the center of that remodeling. Indeed, other members of the KChIP family not expressed in the myocardium behave as transcriptional repressors, while also maintaining the ability to interact with Kv4 channels (*An et al., 2000*; *Carrión et al., 1999*; *Savignac et al., 2005*; *Gomez-Villafuertes et al., 2005*; *Ronkainen et al., 2011*). Therefore, we sought to identify the existence of cardiac KChIP2 transcriptional activity and its significance in electrical remodeling and arrhythmia susceptibility. Here, we find KChIP2 transcriptionally represses a set of miRNAs known as miR-34b and-34c. Through KChIP2 loss, miR-34b/c are elevated, subsequently targeting other ion channel genes defining $I_{Na}$ and $I_{to}$ densities. Either restoring KChIP2 expression or blocking miR-34b/c activity during cardiac stress reverses this remodeling and completely negates the occurrence of re-entrant arrhythmias. Together, this work unveils a novel, transcriptional mechanism for KChIP2, and defines it as a central mediator of cardiac electrical activity.

## Results

### KChIP2 as a transcriptional repressor of miRNAs

This study was approached with the knowledge that acute KChIP2 loss affected the *SCN5A* (Nav1.5), *SCN1B* (Nav$\beta$1), and *KCND3* (Kv4.3) genes in a manner suggesting miRNA activity (*Deschênes et al., 2008*). We therefore performed a miRNA microarray following KChIP2 silencing in neonatal rat ventricular myocytes (NRVMs), resulting in the induction of a number of miRNAs (*Figure 1A*). We evaluated the miRNAs that achieved at least a two fold increase (*Figure 1B*) using TargetScan 7.1 (*Lewis et al., 2005*) to identify potential targeting to the mRNAs *SCN5A*, *SCN1B*, and *KCND3*. Ultimately, we identified miR-34b and −34c as the only miRNAs predicted to target not just one of these ion channel genes, but notably target all three collectively (*Figure 1C*). Notably, we also observed 14 miRNAs decreased greater than two fold (*Figure 1B*). However, a loss in miRNA expression is not consistent with the role of KChIP2 as a transcriptional repressor, and also would not lead to a decrease in ion channel mRNA expression. Real-time qPCR was used to confirm the array results, showing elevation in the mature transcripts for miR-34b and −34c (*Figure 1D*). Importantly, we also performed overexpression of three different cardiac KChIP2 isoforms which

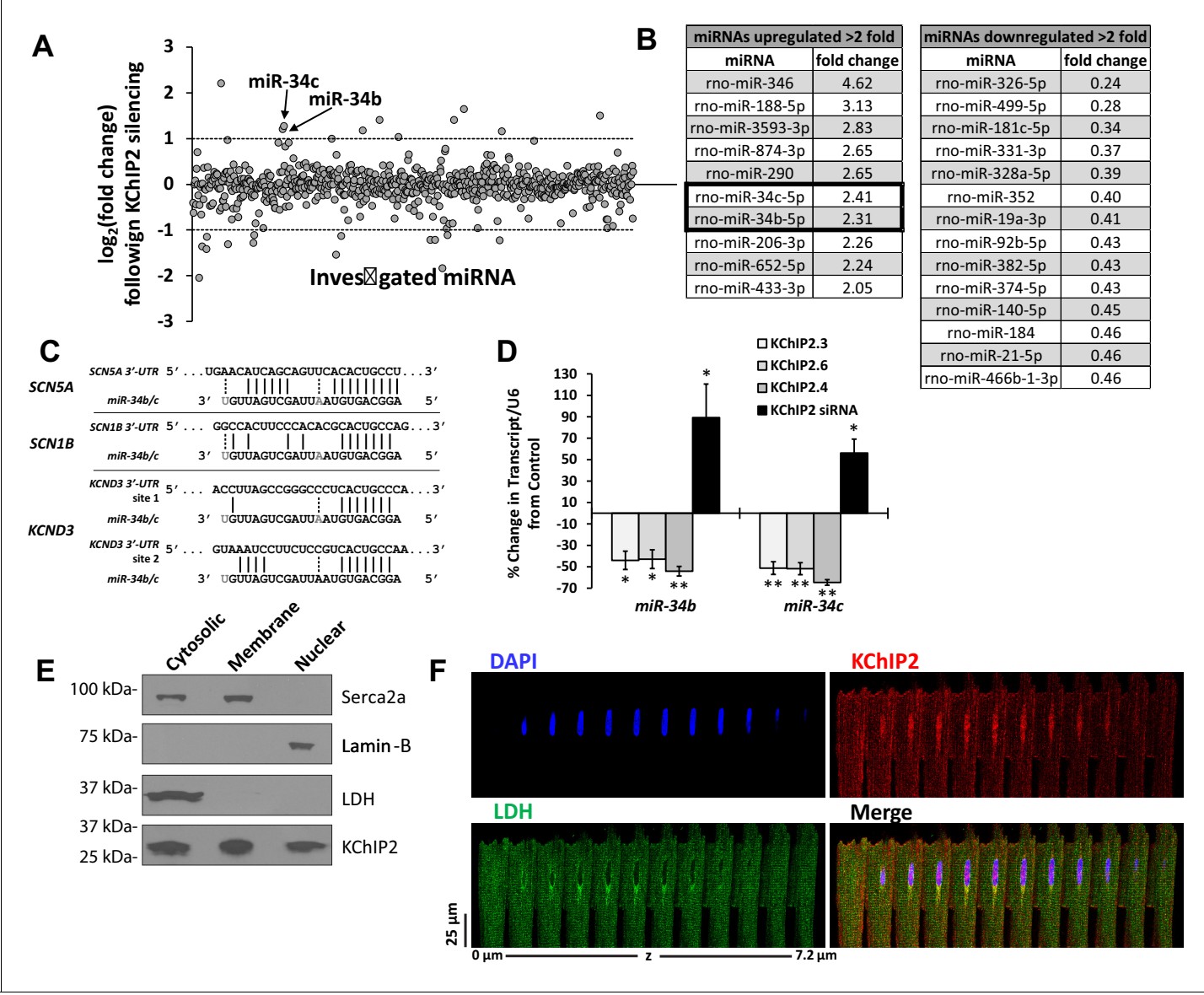

**Figure 1.** miR-34 regulation linked to changes in KChIP2 expression. (A) Results of miRNA microarray showing the log₂ of the fold changes in miR expression following 72 hr of KChIP2 siRNA treatment. Arrow identifies miR-34b and −34c amongst the panel of altered miRNAs. Analysis of miRNAs for mRNA targets using TargetScan 7.1 was restricted to those above two fold induction (dashed line) (B) Tables showing the list of those miRNAs showing at least a two fold increase or decrease following KChIP2 silencing. (C) Alignment of the 3′-UTR of SCN5A, SCN1B, and KCND3 genes with miRs-34b/c from rat, showing hybridization of the seed region. Grayed letters indicate variation in sequence between miR-34b and −34c. A single site of interaction is indicated for SCN5A and SCN1B while two sites exist for KCND3. (D) Real-time qPCR analysis showing percent change of miR-34b/c expression from control cells in NRVM transfected with KChIP2.3 (n = 5), KChIP2.6 (n = 6), KChIP2.4 (n = 4), or KChIP2 siRNA (n = 4–5). (E) Cytosolic, membrane, and nuclear fractions of native adult rat heart tissue. KChIP2 nuclear localization was assessed by using lactate dehydrogenase (LDH), Serca2a, and Lamin-B as cytoplasmic, membrane, and nuclear markers respectively. (F) Representative z-stack images of adult rat ventricular myocyte. Nuclear stained regions (DAPI, blue) show the absence of cytosolic protein LDH (green), while KChIP2 (red) staining reveals significant colocalization. Data presented as mean ± SEM. *p<0.05; **p<0.01, compared to control.

reduced the expression of miRs-34b/c (*Figure 1D*). Together, these changes are consistent with the novel idea that KChIP2 behaves as a transcriptional repressor.

Because KChIP2 is dominantly known as a modulator of Kv4 channels, with cytoplasmic localization (*Takimoto et al., 2002*), we addressed whether it could also localize to the nucleus where it could act as a transcriptional regulator. Indeed, fractionation of adult rat cardiomyocytes into nuclear

fractions reveals endogenous KChIP2 nuclear expression in the absence of contaminating cytosolic (lactate dehydrogenase) and membrane associated proteins (Serca2a) (*Figure 1E*). This is reinforced in the localization patterns of adult cardiomyocytes, showing marked endogenous KChIP2 colocalization in the nucleus in the absence of the cytosolic marker lactate dehydrogenase (*Figure 1F*).

## KChIP2 interaction with the miR-34b/c promoter

To assess whether the transcriptional changes seen in miR-34b/c were the consequence of KChIP2 activity on the promoter, a luciferase assay was conducted containing the cloned minimal miR-34b/c promoter in the presence of KChIP2. Notably, both miR-34b and −34c are transcribed in tandem under the regulation of a shared, intergenic promoter (*Toyota et al., 2008*). To identify potential DNA binding locations for KChIP2, we borrowed from what is known about the putative nucleotide binding sequence for the transcriptional repressor DREAM (KChIP3). This member of the KChIP family shares a high degree of homology with KChIP2, but more importantly has known transcriptional activity occurring through interaction with a nucleotide sequence known as the DRE motif (*Carrión et al., 1999*). MatInspector software (*Cartharius et al., 2005*) was used to evaluate the miR-34b/c promoter for occurrences of this motif, revealing a potential site beginning 254 bp upstream of the miR-34b stem-loop (*Figure 2A*). A region of the promoter 500 bp to 191 bp upstream of the miR-34b stem-loop was cloned into the pGL4.10 luciferase vector and co-transfected with several KChIP2 isoforms into cos-7 cells. When compared to a GFP transfected control without KChIP2, we observed significant repression in the presence of KChIP2.3, 2.4, and 2.6 (*Figure 2A*), showing that KChIP2 can directly act on the miR-34b/c promoter to impart repressive action. To determine if physical KChIP2 interaction with the promoter mediates the repressive state, native adult rat cardiomyocytes were used to perform chromatin immunoprecipitation, followed by qPCR with a primer set flanking the identified DRE site. KChIP2 pull-down resulted in significant enrichment of the DRE containing PCR fragment when compared to an IgG control (*Figure 2B*).

To identify if the DRE site within the promoter fragment is responsible for the repression caused by KChIP2, the core nucleotide sequence was deleted from the promoter (*Figure 2C*). This attenuated the repressive action of KChIP2, implying that KChIP2 is capable of recognizing the same putative DNA binding motifs as DREAM and uses it to induce repressive action. Additionally, it is known that transcriptional derepression of DREAM is regulated through $Ca^{2+}$ binding to EF-hand motifs (*Carrión et al., 1999*). Therefore, to further characterize KChIP2 activity, the reporter assay was conducted following incubation with 10 mM caffeine to induce global elevations in $Ca^{2+}$. This led to significant activation of the promoter (*Figure 2D*), reinforcing the transcriptionally repressive nature of KChIP2 and its conserved mechanisms with DREAM. Together, this data demonstrates that KChIP2 behaves as a transcriptional repressor on the promoter of miR-34b/c by direct binding to the putative DRE motif.

## SCN5A, SCN1B, and KCND3 targeted by miR-34b/c

Previous studies identified reduction in Nav1.5, Navβ1, and Kv4.3 following KChIP2 silencing (*Deschênes et al., 2008*). Having observed that KChIP2 knock-down elevates miR-34b/c, we next sought to determine whether miR-34b/c targets these ion channel transcripts to mediate their loss in expression. Precursor miRNAs for miRs-34b/c were transfected into NRVMs to directly elevate their expression. Assessment of the resulting transcripts showed reduced mRNA for *SCN5A* and *SCN1B* following miR-34 expression, compared to a non-targeting control miR (*Figure 3A*). While *KCND3* levels remained unchanged (*Figure 3A*), Kv4.3 protein experienced significant reduction that reinforces the miRNA mode of translational inhibition without mRNA degradation previously noted (*Deschênes et al., 2008*) (*Figure 3B and C*).

To determine if the changes in channel expression was the consequence of miR-34 targeting to the 3′-UTR of these genes, and not the result of an indirect pathway, fragments of the 3′ region containing the seed sequence were fused to the end of a luciferase reporter construct. This construct was co-expressed with the miR-34b/c precursors in HEK293 cells, resulting in reduced activity in all three constructs when compared to a control miR-precursor (*Figure 3E*). Subsequently, mutations were made within the seed region where miR-34 targeting is predicted to bind (*Figure 3D*), which significantly attenuated the repressive action (*Figure 3E*). This suggests that miR-34b/c are indeed

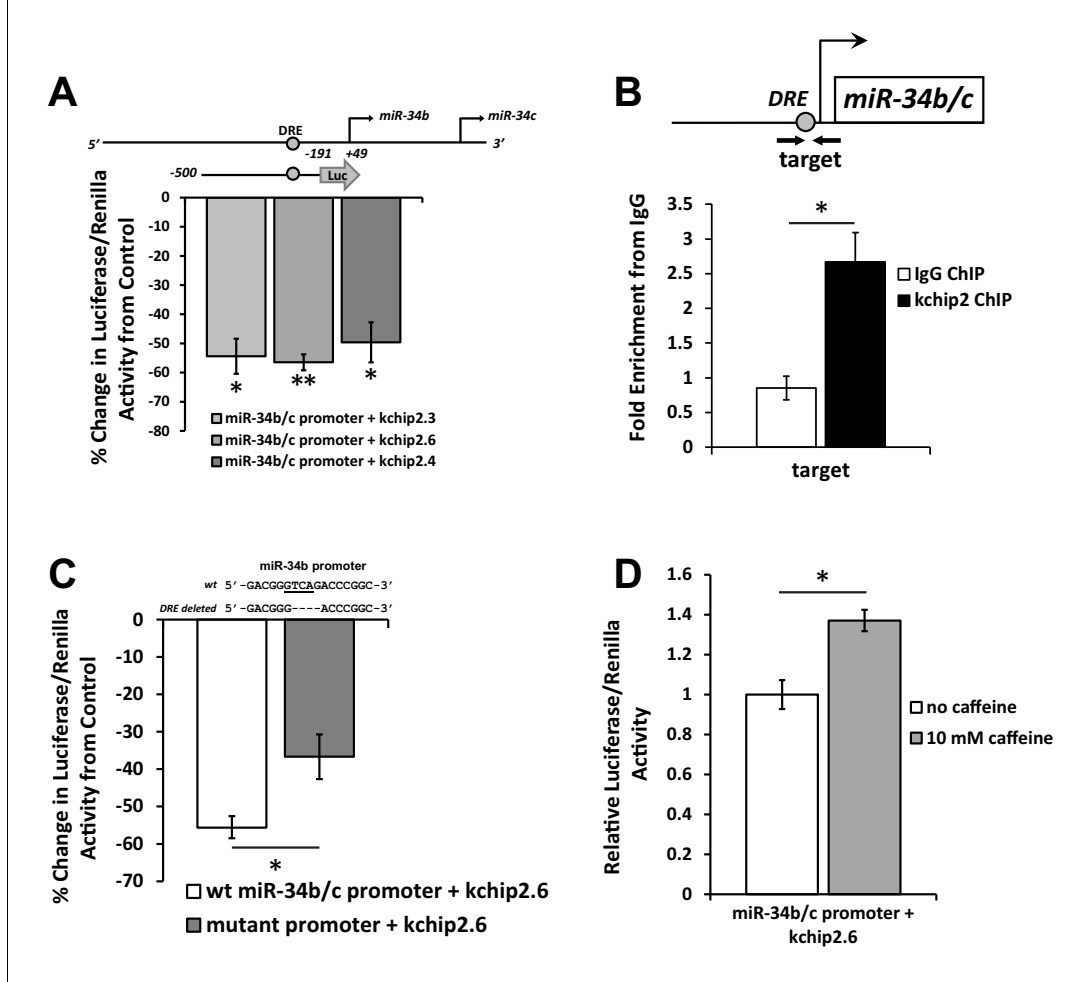

**Figure 2.** KChIP2 represses miR-34b/c expression by direct interaction with a putative DRE motif in promoter. (**A**) A region from −500 to −191 of the miR-34b/c promoter was cloned into the promoterless luciferase construct, pGL4.10. This construct was co-transfected into COS-7 cells in the presence of KChIP2.3 (n = 3), KChIP2.6 (n = 8), or KChIP2.3 (n = 3) and compared to GFP alone. Renillin (pGL4.74) was used as a normalization control. Results are depicted as a % change in activity compared to GFP alone. (**B**) IgG and KChIP2 ChIP-PCR conducted on native adult rat cardiomyocytes. The target primer site residing within the cloned promoter was evaluated for enrichment following pull down (n = 3), showing significant enrichment of the target region. (**C**) Luciferase assay conducted in COS-7 cells to evaluate the outcome of deleting the putative DRE site in the miR-34b/c promoter. COS-7 cells were transfected with the same WT reporter construct inserted into the pGL4.10 vector or with the DRE motif deleted, both in the presence of KChIP2.6. Activity was normalized to renillin (pGL4.74). Deletion of a putative KChIP2 interaction site (DRE motif) partially abolished the repressive effect KChIP2.6 had over the miR-34b/c promoter (n = 4) compared to WT (n = 9). (**D**) COS-7 cells transfected with KChIP2.6 and the pGL4.10 containing the WT miR-34b/c promoter were treated with or without 10 mM caffeine for 6 hr, leading to promoter activation (n = 4). Results were normalized to renillin activity. Data presented as mean ± SEM. *p<0.05; **p<0.01, as indicated or compared to control.

targeting the predicted seed region in the *SCN5A*, *SCN1B*, and *KCND3* genes and directly influencing their expression.

## miR-34b/c functionally regulates $I_{Na}$ and $I_{to}$ density

Functional assessment of changes to $I_{Na}$ and $I_{to}$ were determined through patch clamp recordings in NRVM. Reflecting the changes in mRNA and protein, expression of miR-34b/c precursor produced a significant decline in $I_{Na}$ (*Figure 3F*). $I_{to}$, however, while having trended reductions, did not produce significant loss despite the loss in Kv4.3 protein levels (*Figure 3G*). This can be attributed to a number of reasons. The current evaluation was conducted in rodent myocytes, where $I_{to}$ is comprised of the shared alpha subunits Kv4.2 and Kv4.3, which comprise a fast component of $I_{to}$ referred to as $I_{to,f}$. Additionally, there are the contributions of Kv1.4, another potassium channel subunit, which

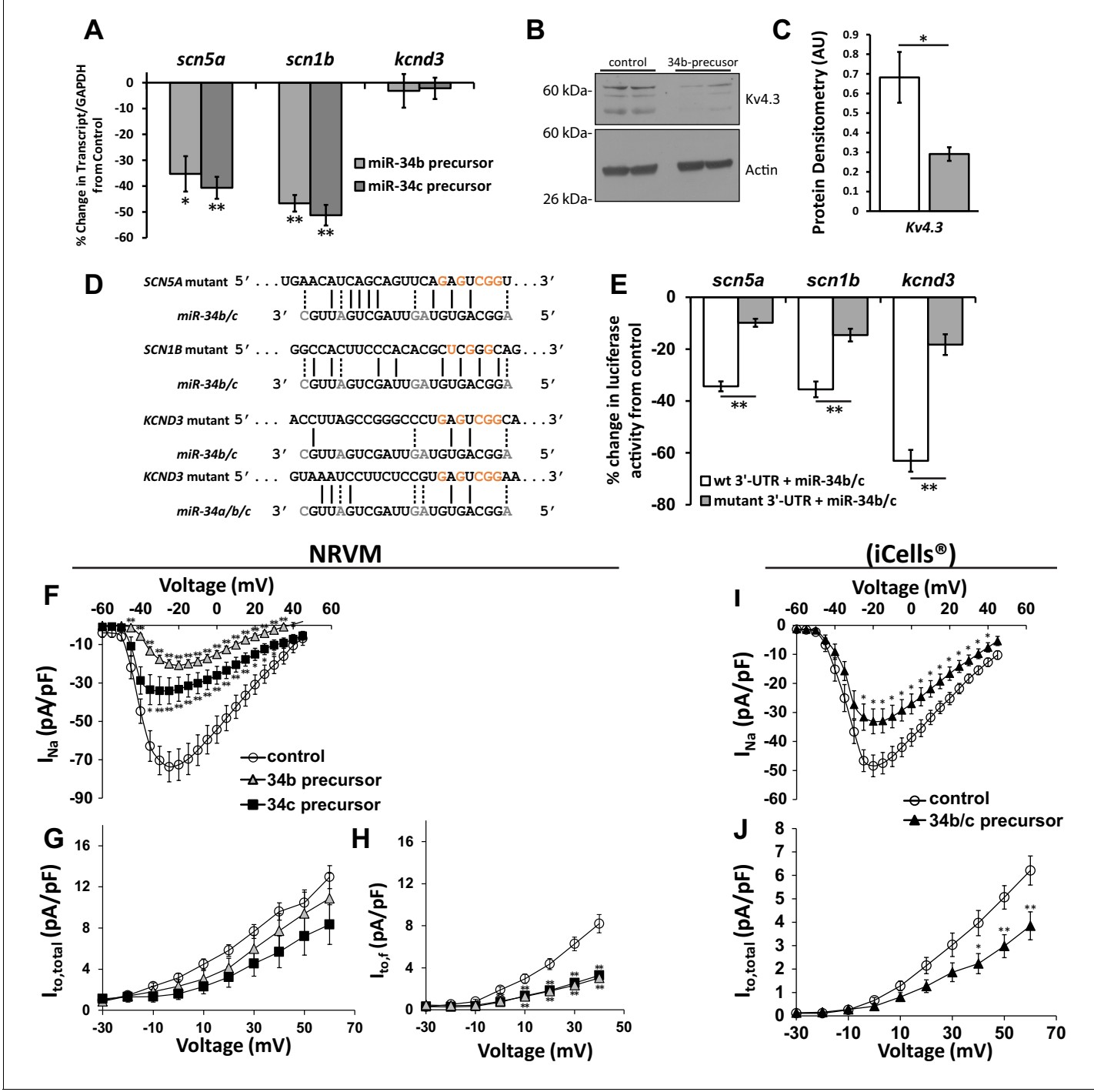

**Figure 3.** Cardiac ion channel directly regulated by miR-34a/b/c through interaction with their 3'-UTR. (A) NRVM over-expressing precursors for miR-34b/c were collected for mRNA transcript levels. Results (normalized to non-targeting miR) show down-regulation of SCN5A, and SCN1B, but unchanged levels for KCND3 (n = 7–8). (B) Protein levels from NRVM with over-expressed miR-34b showing reduced protein expression for Kv4.3 (KCND3). Multiple bands for Kv4.3 represent different glycosylation states of the protein. (C) Summary data of the immunoblot of Kv4.3 (n = 4). (D) Alignment of the 3'-UTR of SCN5A, SCN1B, and KCND3 genes with miRs-34b/c, with mutations made to the seed regions (highlighted in red) to disrupt interaction at the seed region. (E) Reporter assay with the 3'-UTR cloned into pmiRGlo reporter construct. Luciferase activity in HEK cells transfected with WT or mutant 3'-UTRs. Results are presented as a percent change from a non-targeting miR precursor (n = 5) normalized to renillin activity. (F) I/V curves for $I_{Na}$ measured in NRVM over-expressing precursors for control (n = 24), miR-34b (n = 24), or miR-34c (n = 18). (G) I/V curves for $I_{to,total}$ measured in NRVM over-expressing precursors for control (n = 15), miR-34b (n = 12), or miR-34c (n = 11). (H) $I_{to,f}$ was also assessed in NRVM through kinetic subtraction of $I_{to,s}$. Resulting I/V curves now reveal a significant reduction in current density in miR-34b (n = 14) and miR-34c (n = 15)

*Figure 3 continued on next page*

*Figure 3 continued*

precursor treated cells compared to control (n = 16). (I) The same experiments conducted in human derived cardiomyocytes (iCells) expressing either control or miR-34b/c precursor together, measuring $I_{Na}$ (control, n = 24; miR-34b/c, n = 21) and (J) $I_{to,total}$ (control, n = 24; miR-34b/c, n = 25). Data presented as mean ± SEM. *p<0.05; **p<0.01, as indicated or compared to control. See also *Figure 3—figure supplement 1*.

The following figure supplement is available for figure 3:

**Figure supplement 1.** Kv4.2 (kcnd2) expression in NRVM following expression of miR-34b/c precursor.

encodes a slow component, referred to as $I_{to,s}$. These descriptions are attributed to the respective rates of recovery from inactivation for each of these channels (*Niwa and Nerbonne, 2010*). Notably, our patch protocol in *Figure 3G* took into account the contributions of all three subunits, or $I_{to,total}$. Therefore, despite reductions in Kv4.3 protein expression the change in current resists as it is not the predominant channel contributing to $I_{to}$. Importantly, Kv4.2 and Kv1.4 do not contain a miR-34 seed region. In fact, in response to miR-34b/c expression, mRNA levels for Kv4.2 actually experienced a trended elevation (*Figure 3—figure supplement 1*) which could also contribute to the lack of reduction in $I_{to}$. We therefore modified our patch protocol to probe just $I_{to,f}$ and remove the contribution of Kv1.4, which now revealed a significant reduction in the $I_{to,f}$ density (*Figure 3H*). To further identify if the presence of Kv4.2, Kv4.3, and Kv1.4 in rats could explain the resisted change in $I_{to}$, cardiomyocytes derived from human induced pluripotent stem cells (iCells) were used, since in the human background, Kv4.3 is the dominant contributor to $I_{to}$ (*Niwa and Nerbonne, 2010*). Expression of miR-34b/c precursors now produced a significant loss in $I_{to}$ density (*Figure 3J*), while also maintaining reductions in $I_{Na}$ (*Figure 3I*). Importantly, this not only satisfies why $I_{to}$ loss was resistant in the NRVMs, but identifies conservation of miR-34 activity across species, implicating the importance of miR-34s in human cardiac ion channel regulation.

## KChIP2 regulates miR-34b/c expression during cardiac stress

To begin to understand the pathogenic importance of this pathway, NRVMs were cultured in 100 μM phenylephrine (PE) for 48 hr to mimic neuro-hormonal overload in a stressed myocardium. PE stimulation resulted in a dramatic decline of *KCNIP2* (KChIP2), while also yielding significant elevation in miR-34b/c (*Figure 4A and B*). These conditions resulted in reductions in expression for *SCN5A*, *SCN1B*, and *KCND3* transcripts (*Figure 4C*). Critically, maintaining KChIP2 levels through use of adenovirus encoding KChIP2 (Ad.KChIP2) normalized the expression of miRs-34b/c while reversing the loss in *SCN5A* and *SCN1B*; however, *KCND3* levels remained suppressed (*Figure 4B and C*). Functional evaluation on both $I_{Na}$ and $I_{to,f}$ shows significant loss in density following PE treatment (*Figure 4D and E*), reflecting the changes we see in transcript expression and mimicking ion channel remodeling observed in HF. However, Ad.KChIP2 treatment restored the current density for both currents, despite *KCND3* transcript expression being unaffected by KChIP2 expression. These observations strongly implicate a role for KChIP2 in maintaining proper electrical expression during pathological remodeling in the stressed heart. Moreover, we were able to observe significant reduction of KChIP2 and elevation of miR-34b/c within failing human heart tissue compared to non-failing (*Figure 5A*). Reinforcing this conservation was the identification of a predicted DRE motif proximal to the transcriptional start site, in the human miR-34b/c promoter as evaluated by MatInspector (*Figure 5B*). At the same time, significant loss of *SCN5A* and *KCND3* transcripts in failing tissue (*Figure 5C*) also show conservation of miR-34b/c targeting within their 3'-UTRs (*Figure 5D*). Interestingly, *SCN1B* does not preserve its target site in humans, however, we also observed no significant reduction in transcript expression from failing heart tissue (*Figure 5C*). Together, this reinforces the concept of KChIP2 as a core transcriptional regulator of electrical activity under normal and pathologic conditions.

To address the specific activity of miR-34b/c in mediating these changes in ion channel expression, NRVMs and iCells were transfected with miR-34b/c antimir molecules during the duration of PE treatment. Much like KChIP2 delivery which reduced miR-34b/c expression, directly blocking miR-34b/c activity maintained $I_{Na}$ in both rat and iCells (*Figure 6A and D*), further implicating miR-34b/c in the direct regulation of these ion channel transcripts. However, $I_{to,total}$ density in the NRVMs did not observe the same rescue (*Figure 6B*). We believe this is once again explained by the

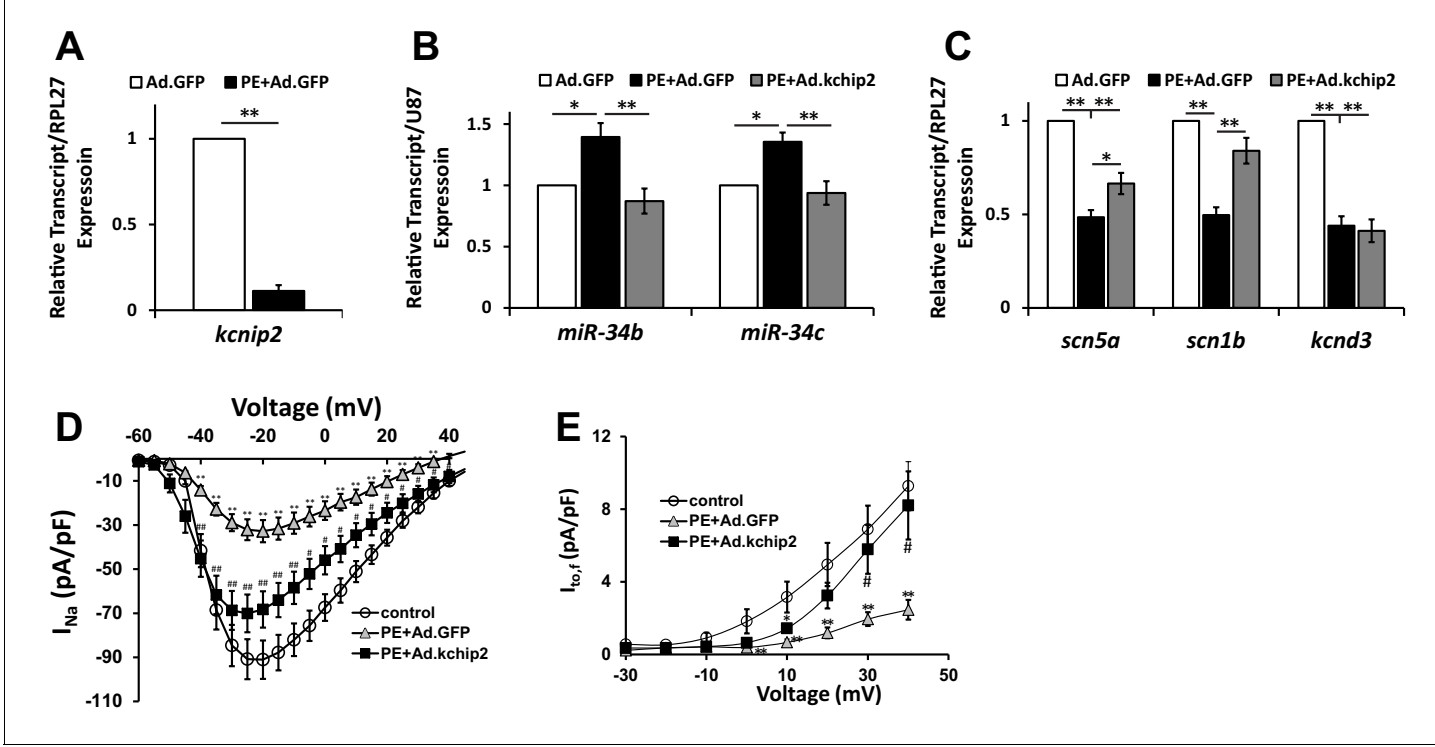

**Figure 4.** In vitro cardiac disease signaling links KChIP2 loss with miR-34 elevation. (**A**) Real-time qPCR evaluation of relative kcnip2 following treatment with 100 μM PE for 48 hr in NRVM (n = 6). Results normalized to ribosomal protein RPL27. (**B**) Evaluation of miR-34b (n = 8) and miR-34c (n = 7) relative expression in NRVM under control (no PE with Ad.GFP), 100 μM PE with Ad.GFP, or 100 μM PE with Ad.KChIP2 to maintain KChIP2 expression during the 48 hr treatment. Expression levels were normalized to small nucleolar RNA, U87. (**C**) The same treatment conditions in (**B**), evaluating relative mRNA expression for SCN5A (n = 10), SCN1B (n = 10), and KCND3 (n = 7). (**D**) Functional current-voltage measurements of $I_{Na}$ from NRVM under control (n = 29), PE+Ad.GFP (n = 27), and PE+Ad.KChIP2 (n = 30). (**E**) I/V curve for $I_{to,f}$ recordings in control (n = 7), PE+Ad.GFP (n = 9) and PE+Ad.KChIP2 (n = 9). Data presented as mean ± SEM. *p<0.05, **p<0.01, as indicated or compared to control, #p<0.05, compared to PE+Ad.GFP.

contributions of Kv1.4 and Kv4.2, in addition to Kv4.3 in defining rodent $I_{to}$. In fact, by probing just $I_{to,f}$, we revealed a significant, but incomplete restoration following miR-34b/c block (*Figure 6C*). Notably, the same experiment conducted in iCells where Kv4.3 is the dominant subunit, resulted in the full restoration of $I_{to}$ (*Figure 6E*). To be more certain the restoration of current density was specific to miR-34b/c targeting the underlying subunits encoding $I_{Na}$ and $I_{to}$, rather than a general rescue in the molecular state of the cell, the repolarizing current $I_{Kr}$ was assessed in iCells. PE successfully reduced this current, which is known to be reduced by cardiac stressors, however, it was unable to be rescued by miR-34b/c block (*Figure 6—figure supplement 1*). Critically, this shows that KChIP2 regulation of $I_{Na}$ and $I_{to}$ is enacted through specific targeting of miR-34b/c activity, while the use of iCells displays mechanistic conservation in human derived cells.

## Inhibition of miR-34b/c blocks arrhythmia induction

Dysregulation of $I_{Na}$ and $I_{to}$ have been previously associated with arrhythmogenesis (*Starmer et al., 2003*; *Kuo et al., 2001*). Therefore, in order to test the consequence of $I_{Na}$ and $I_{to}$ loss and the involvement of miR-34b/c in regulating susceptibility to arrhythmic events, optical mapping was performed in NRVM monolayers. As before, cells were exposed to 100 μM PE for 48 hr following treatment with either a control or miR-34b/c antimir. Using point stimulation, we submitted the monolayers to baseline pacing (S1) followed by a single premature stimulus (S2) over a range of S1-S2 coupling intervals. Immediately following S2 capture, the occurrence of rapid, non-paced activity (arrhythmia) was assessed.

*Figure 7A* shows representative activation maps during S1 (top) and S2 (bottom) pacing. In all conditions, activation during S1 pacing shows uniform wavefront propagation, with evidence of

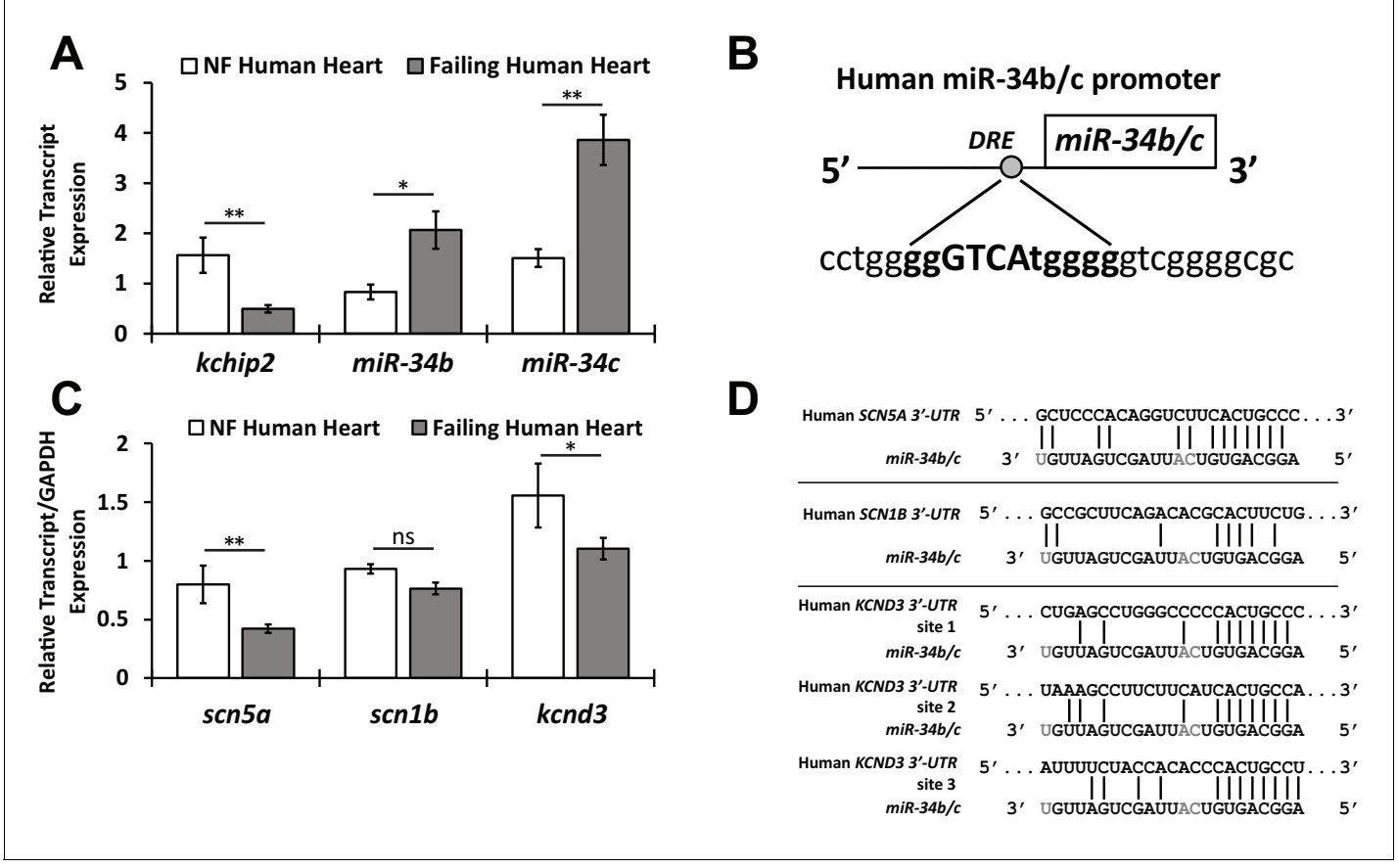

**Figure 5.** Preservation of the KChIP2/miR-34b/c axis in human heart failure. (A) Human tissue taken from the left ventricle of non-failing (NF) (n = 8) and failing patients (n = 20) evaluating KChIP2 and miR-34b/c RNA expression. KChIP2 levels were normalized to GAPDH and miR expression to small nucleolar RNA U6. (B) Evaluation of the human miR-34b/c reveals a conserved DRE motif in proximity of the miR-34b stem loop (−242 bp), as predicted by MatInspector, suggesting conservation of KChIP2 activity in the regulation of miR-34b/c expression. (C) Human heart failure tissue evaluating RNA levels for *SCN5A*, *SCN1B*, and *KCND3*. Significant reductions in heart failure samples (n = 20) were observed for *SCN5A* and *KCND3*, but not for *SCN1B*, compared to non-failing tissue (n = 8). (D) Alignment of the 3'-UTR of SCN5A, SCN1B, and KCND3 genes with miRs-34b/c from human. Grayed letters indicate variation in sequence between miR-34b and −34c. A single site of interaction is indicated for SCN5A, matching observations in the rat, while *KCND3* has three potential sites, compared to two observed in the rat. Notably, SCN1B miR-34b/c targeting is not conserved in human shown by imperfect hybridization in the seed region. Data presented as mean ± SEM. *p<0.05; **p<0.01, as indicated or compared to control. #p<0.05, ##*PP*<0.01 compared to PE+Ad.GFP.

conduction slowing following PE + control antimirs, consistent with reduced $I_{Na}$ density. Compared to S1 pacing, propagation during S2 pacing was slower in all conditions; however, in PE + control antimirs significant impulse slowing (isochrone crowding) and block (solid line) were observed. Critically, this block was sufficient to cause sustained reentrant excitation in 5 of 7 monolayers (*Figure 7B and C*). Remarkably, PE + miR-34b/c inhibition prevented conduction block and mitigated conduction slowing, protecting all monolayers from sustained re-entry (*Figure 7C*).

To determine the electrophysiological substrate responsible for the reentrant activity observed, monolayers were evaluated for changes in APD and conduction velocity. Reflecting the changes in $I_{Na}$ and $I_{to}$ expression, exposure to PE for 48 hr significantly prolonged APD and slowed conduction velocity compared to control dishes across multiple pacing cycle lengths (*Figure 7D and E*). APD prolongation from PE was unresponsive to miR-34b/c inhibition; however, this was anticipated as we previously determined $I_{to}$ is not restored by miR-34b/c block in NRVM due to the additional Kv1.4 and Kv4.2 mediated current. However, treatment with miR-34b/c antimir, which maintained $I_{Na}$ density in isolated myocytes (*Figure 6A*), produced a trend towards restoration of conduction velocity,

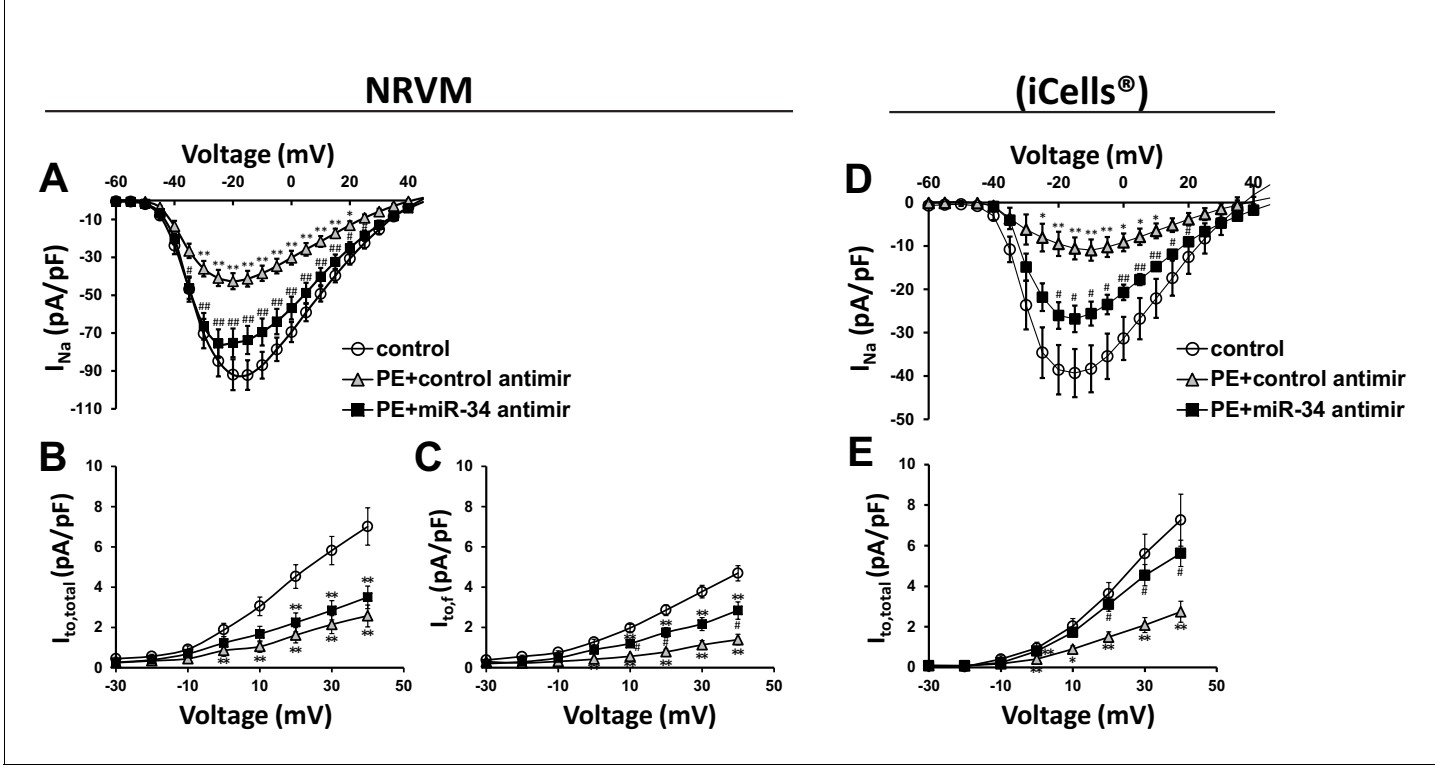

**Figure 6.** miR-34 block reverses loss of both $I_{Na}$ and $I_{to}$ in disease signaling. (**A**) $I_{Na}$ I/V curve measured in NRVM transfected with either non-targeting antimirs (control, n = 26), non-targeting miR + 100 µM PE (PE+control antimir, n = 20), or miR-34b/c antimirs + 100 µM PE (PE+miR-34 antimir, n = 21) for 48 hr. (**B**) $I_{to,total}$ I/V measurements in NRVM showing current density is lost in PE+control (n = 16) and remains down in the PE+miR-34 antimir (n = 16), compared to control (n = 17) cells. (**C**) $I_{to,f}$ I/V measurements in NRVM. Cells treated with PE+control antimir (n = 22) have reduced current density, that is now partially restored in the PE+miR-34 antimir (n = 23) cells compared to control (n = 27). (**D**) I/V curve for $I_{Na}$ taken in iCells showing that miR-34 antimirs (n = 6) can rescue current density back toward control (n = 6), when compared to PE+control (n = 6). (**D**) I/V curve for $I_{to,total}$ measurements in iCells showing miR-34b/c antimir in the presence of PE (n = 15) can rescue current density towards control (n = 15) while PE+control (n = 15) remains reduced. Data presented as mean ± SEM. *p<0.05 versus control, **p<0.01, as indicated or compared to control antimir, #p<0.05, ##p<0.01 compared to PE+control antimir. See also *Figure 6—figure supplement 1*.

The following figure supplement is available for figure 6:

**Figure supplement 1.** $I_{Kr}$ is insensitive to miR-34 block following PE stimulation.

suggesting other mechanisms of conduction slowing following PE treatment that are uninfluenced by miR-34b/c activity.

Therefore, to more precisely assess changes in cellular excitability, we determined the effective refractory period (ERP) under each condition. Reflecting the prolonged APD and reduced $I_{Na}$, PE treated cells displayed a significantly longer ERP (*Figure 7F*) than control cells. However, treatment with the miR-34 antimir significantly shortened ERP towards control. Notably, this recovery occurred in the absence of a shortened APD, suggesting a significant recovery of $I_{Na}$ excitability. Thus, even without being able to rescue $I_{to}$, we were still able to restore cellular excitability through miR-34b/c inhibition and limit the occurrence of conduction block and reentry. Overall, the observation KChIP2 can normalize electrical remodeling in a setting of myocardial stress highlights a much expanded and multimodal role in establishing the cardiac electrical state.

## Discussion

This study established a novel transcriptional role for cardiac KChIP2, whereby it maintains a repressive influence over the miR-34b/c promoter. KChIP2 loss either by direct silencing or pathologic

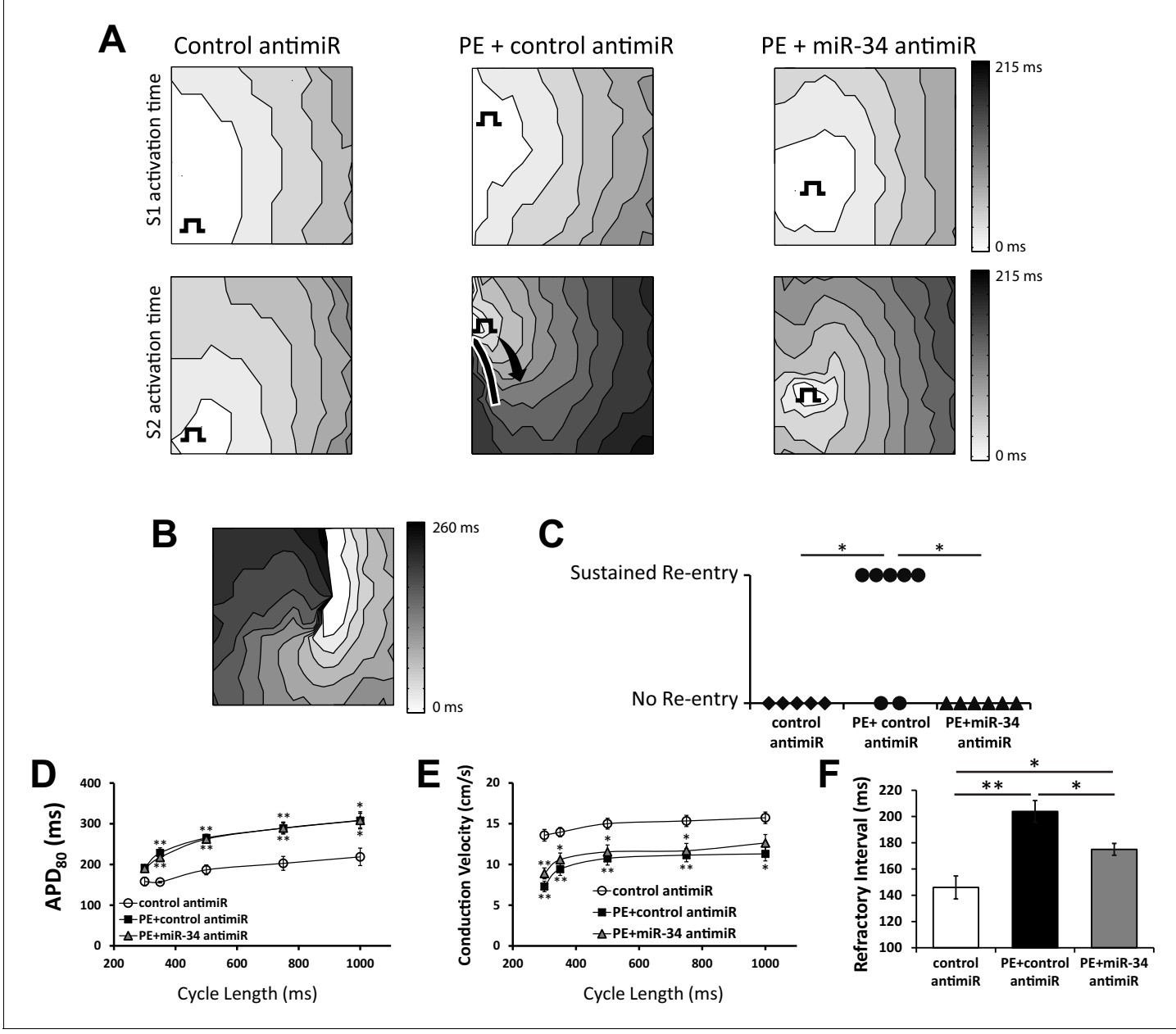

**Figure 7.** miR-34 block retains excitability in NRVM monolayers following prolonged PE treatment. (**A**) Isochronal conduction maps of monolayers submitted to PE (100 μM) with either a non-targeting control or miR-34b/c antimir. Conduction maps on the top row represent the final S1 (750 ms) preceding the S2, showing no pre-existing abnormalities in propagation. The square function represents the site of pacing. The second row shows the first incidence of capture of the premature stimulus (S2). PE + control antimir results in significant conduction block around the pacing site (solid line). Conduction block was minimal in control and PE + miR-34b/c antimir groups. (**B**) Conduction map showing an example of sustained reentry for the PE + control antimir treated group shown in (**A**). (**C**) Summary data for the occurrence of sustained reentry following S1S2 pacing. (**D**) Restitution curve of $APD_{80}$ in paced NRVM monolayers treated with either control antimir (n = 6–8), PE + control antimir (n = 6–11), or PE + miR-34b/c antimir (n = 7–12). (**E**) Conduction velocity restitution curve in paced NRVM monolayers treated with either control antimir (n = 6–8), PE + control antimir (n = 6–11), or PE + miR-34b/c antimir (n = 17–12). (**F**) Measurement of the effected refractory interval evaluated by identifying the shortest premature stimulus that would elicit capture or arrhythmia induction, under control (n = 6), PE+control antimir (n = 13), and PE+miR-34b/c antimir (n = 12). Data presented as mean ± SEM. *p<0.05, **p<0.01, as indicated or compared to control antimir.

means, removes repression over miR-34b/c expression. Consequentially, reductions in transcript and protein expression for Nav1.5, Navβ1, and Kv4.3 are observed as an outcome of miR-34b/c targeting to seed regions present in the 3'-UTR of these genes, allowing KChIP2 to manipulate functional expression of a host of critical cardiac ion channel genes, ultimately acting as a key regulator of cardiac excitability and arrhythmia susceptibility.

While we evaluated a discrete pathway targeted by KChIP2 transcriptional activity, there are doubtless many other gene targets. To begin to address this discussion, a gene expression array was performed on NRVM following 48 hr of KChIP2 silencing. Evaluation of genes that experienced at least a two fold change revealed an increase in expression for 293 genes and a decrease in expression for 407 genes in response to KChIP2 silencing (*Supplementary file 1*). Notably, of the genes experiencing increased expression, 192 of them (65.5%) were predicted to contain a DRE motif within promoter elements, implicating the potential of KChIP2 transcriptional activity directly mediating these changes. Additionally, of the genes responding with reduced expression, 71 of them (17.4%) were predicted to contain a miR-34b/c target site within their 3'-UTR. Importantly, we see significant reduction to *SCN5A* and *SCN1B*, consistent with previous data and the mechanisms investigated here. Considerably, whether these changes are the direct consequence of KChIP2 transcriptional repression, or more indirect KChIP2 dependent mechanisms, including prolonged APD from loss in $I_{to}$ density contributing to altered $Ca^{2+}$ handling, the results are still relevant to cardiac remodeling, particularly given the associated loss of KChIP2 in cardiac disease states. Notably, a diverse range of gene pathways were implicated in response to KChIP2 loss, including cardiovascular signaling, regulation to G-protein coupled receptor pathways, relaxation and contraction, TGFβ signaling, apoptotic, and NFκB dependent signaling mechanisms, all of which have a relevance in disease remodeling (*Supplementary file 1*). Importantly, these data are supportive of our conclusion that KChIP2 is a key regulator of cardiac pathology. While our study focused on the transcriptional pathway by which KChIP2 could exert a concerted regulation of $I_{Na}$ and $I_{to}$, further investigations pursuing some of the targets highlighted from this gene expression array will likely reveal a broader role for KChIP2 as a transcriptional regulator of cardiac physiology much like is seen for KChIP3 (DREAM) in the brain.

Indeed, the role of KChIP2 as a multimodal regulator of cardiac ion channels has been an emerging topic. Recent work has identified KChIP2 regulation of $Ca_v1.2$ through direct interaction with an inhibitory N-terminal domain on the channel, effectively reducing $I_{Ca,L}$ in the absence of KChIP2 (*Thomsen et al., 2009*; *Foeger et al., 2013*). Our previous work has also suggested that KChIP2 is part of a larger macromolecular complex that includes subunits for both $Na_v1.5$ and Kv4 channels that lead to functional increases in both currents when coexpressed with KChIP2 (*DeschenesDeschênes et al., 2008*). In the same study we also identified transcriptional changes in *SCN5A* and *SCN1B* following acute knockdown of KChIP2 in NRVM, which provided the motivational basis for the work presented here. Taken together, these observations suggest a highly promiscuous nature for KChIP2.

Notably, another member of the KChIP family, KChIP3, has also been discovered to interact with multiple membrane proteins, including regulation of Kv4 channels (*Mellström et al., 2008*; *Buxbaum et al., 1998*), while also displaying $Ca^{2+}$ regulated transcriptional repression (*Carrión et al., 1999*). Even the role of KChIP3 as a transcriptional repressor is multimodal, including direct binding to DRE motifs, in addition to interacting with and suppressing the activity of the cAMP response element-binding protein (CREB), an established transcriptional activator (*Ledo et al., 2002*). Both of these processes are $Ca^{2+}$ regulated, due to three functional high affinity EF-hand motifs residing in the protein (*Carrión et al., 1999*). Occupancy of these sites upon increases in intracellular $Ca^{2+}$ lead to conformational changes that cause DNA binding release (*Carrión et al., 1999*) or dissociation from CREB (*Ledo et al., 2002*), causing de-repression of downstream gene targets. Given that the entirety of the KChIP gene family displays strong conservation around these EF-hand residues, suggests conservation of these $Ca^{2+}$ regulated responses. Indeed, we observed that caffeine stimulation produced increased activity of the miR-34b/c promoter (*Figure 2D*) in the presence of KChIP2. Additionally, when we deleted the DRE element in the miR-34b/c promoter, we observed an incomplete removal of suppression (*Figure 2C*). However, as KChIP3 represses gene expression through alternative CREB dependent regulation, the partially retained repressive activity may be attributed to this secondary function. Further analysis of the promoter by MatInpsector revealed several potential sites of predicted CREB binding that may have

allowed for partially maintained KChIP2 suppression, even in the absence of the DRE site. Given that KChIP2 and KChIP3 share a high degree of homology only reinforces the observation of multiple activities for KChIP2 as well.

The physiologic implications of KChIP2 targeting miR-34b/c expression is one of tremendous significance for many cardiac pathologic states. Rapid depletion of KChIP2 protein is a widespread event that underlies remodeling in many cardiac diseases, including chronic HF, MI, and atrial fibrillation (*Nattel et al., 2007*). Considerably, these diseases also present with reductions in $I_{to}$ and $I_{Na}$. The relationship between KChIP2 and $I_{to}$ has been heavily studied, frequently identifying that KChIP2 loss induces the destabilization of Kv4.2/4.3 channels and mediates the decline in current density (*Foeger et al., 2013*). However, the work presented here offers the unique alternative that translation block through miRNA interaction mediates the decline in Kv4.3. Given that Kv4.2 does not contain a miR-34 target region in its 3'-UTR, but still experiences degradation following KChIP2 loss, it is likely that both mechanisms contribute to the resulting loss in $I_{to,f}$. However, it is also observed that reduced KChIP2 expression stimulated by phenylephrine + propranolol in in vitro cultures of NRVM experienced increased Kv4.2 protein while KChIP2 and Kv4.3 levels were reduced (*Panama et al., 2011*), supporting the opportunity for miRNA dependent translational block targeting Kv4.3, rather than just destabilization of all Kv4 channels.

In the same settings of cardiac disease where KChIP2 is down, there are also observations of $I_{Na}$ depletion (*Valdivia et al., 2005*; *Zicha et al., 2004*). Our data of miR-34 targeting Na$_v$1.5 provides a means for describing this loss in activity. Notably, others have shown a loss in the full length transcript for Nav1.5 mRNA and a corresponding increase in a truncated isoform without the miR-34b/c target region present (*Shang et al., 2007*), reinforcing the observations for miR-34b/c mediating the decline of *SCN5A*. Overall, the consequential loss of both $I_{Na}$, and $I_{to}$, suggests KChIP2 loss during cardiac stress may be a nodal event in a cascade of gene regulation defining electrical remodeling in the stressed myocardium. Indeed, earlier work was done that sought to determine the significance of KChIP2 in the development of hypertrophic remodeling. In a rat TAC banding model, it was observed that maintaining KChIP2 expression attenuated hypertrophy and pathogenic remodeling that otherwise lead to a worsening myocardium during pressure overload (*Jin et al., 2010*). This reverse in remodeling was attributed to changes in intracellular Ca$^{2+}$ signaling brought on by restoration of an abbreviated APD. Yet, we were able to observe that inhibition of miR-34b/c could also attenuate adverse remodeling without influencing APD (*Figure 7*) implicating multiple pathways of KChIP2 intervention. Indeed, the miR-34 family has recently been implicated in the development and progression of hypertrophy and heart failure, in rodent models of both MI and pressure overload (*Bernardo et al., 2012*). Critically, these studies, combined with our own data, show that blockade of the miR-34 family can attenuate pathologic remodeling, expanding the significance of KChIP2 and miR-34 in cardiac pathogenesis.

There are still some challenges in understanding the role of KChIP2 in the progression of hypertrophy and heart failure. Investigations conducted in KChIP2 null mice have shown that when submitted to TAC banding, there is no worsened phenotype when compared to wild type mice (*Speerschneider et al., 2013*). In fact, arrhythmia susceptibility was lowered in the KChIP2 null mice during heart failure, believed to be the result of reduced dispersion of repolarization. At the same time, there were no observed changes to $I_{Na}$. While our current understanding is unable to account for this disparity, it may be that compensatory regulation exists in these mice as a consequence of constitutive KChIP2 absence during development, fundamentally changing its regulatory significance. Evidence for this is observed when restoring KChIP2 expression in myocytes isolated from KChIP2 null mice, which resulted in no rescue of Kv4.2 protein expression or recovery of $I_{to,f}$ (*Foeger et al., 2013*). However, restoration of KChIP2 following acute loss from pathologic consequences in a rat model was able to rescue $I_{to,f}$ (*Jin et al., 2010*), consistent with what we see in our own maintenance of KChIP2 following prolonged PE exposure (*Figure 4E*). The significance of this begins to suggest deviations in KChIP2 regulatory impact depending on acute versus constitutive loss.

Ultimately, our endpoint was to determine whether electrical dysregulation brought on by KChIP2 loss was able to influence arrhythmia susceptibility through the activity of miR-34b/c. Despite only rescuing $I_{Na}$ and not $I_{to}$ in the NRVMs, as evidenced by the shortened ERP with sustained APD prolongation (*Figure 7D and F*), we found this was sufficient to rescue arrhythmia induction following PE treatment (*Figure 7C*). Indeed, previous studies have revealed the relationship between changes

in Na$^+$ channel density and arrhythmia induction. As $I_{Na}$ becomes compromised, it begins to resolve an expanding interval of premature stimuli declared the vulnerability period. Within this interval, reentry is more likely to occur as a result of non-uniform conduction block surrounding the point of excitation (*Starmer et al., 2003*). Both theoretical (*Starmer et al., 1991*, *1993*) and experimental (*Nesterenko et al., 1992*; *Starmer et al., 1992*) studies show that when Na$^+$ channel availability is reduced, the vulnerable period increases. Therefore, by restoring Na$^+$ channel through miR-34b/c inhibition, we are effectively minimizing the vulnerable period and making unidirectional conduction block less likely to occur.

Care must still be taken before translating these mechanisms to the clinical setting. Our investigated pathway was developed using cultured rodent myocytes, differing from human electrophysiology in its APD and the impact of underlying currents. We must also understand the electrical impact of miR-34 inhibition in vivo. However, we know from this investigation that miR-34b/c are elevated in native human HF tissue (*Figure 5A*), and that functionally, the inhibition of miR-34b/c in human derived cardiomyocytes following stress can achieve restoration of both $I_{Na}$ and $I_{to}$ (*Figure 6C and D*), reinforcing species dependent conservation. At the same time, conduction block due to compromised cellular excitability has long been understood to be important for clinically relevant arrhythmias (*Shah et al., 2005*). These observations together suggest strong therapeutic potential for targeting miR-34 in the treatment of electrical instabilities. Currently, the use of locked nucleic acids and related technologies have been used to successfully target miRNA activity in vivo (*Olson, 2014*). While miR-34b/c is also expressed outside the heart, it is unclear what long-term consequences its inhibition will have as a therapeutic. However, these outcomes will have to be weighed against the potential therapeutic advantage it will have in alleviating cardiac events.

Overall, this newly identified KChIP2/miR-34 pathway reflects electrical remodeling observed within multiple cardiac pathologies. Moreover, the events brought on by KChIP2 loss are critical in initiating electrical instabilities and arrhythmias implicated in sudden cardiac death. The identification of KChIP2 transcriptional capacity significantly reshapes its role in cardiac biology as a core mediator of cardiac electrical activity and reveals KChIP2 and miR-34 as therapeutic targets for managing arrhythmogenesis in heart disease.

## Materials and methods

### Isolation and cell culture of neonatal rat ventricular myocytes and human derived cardiomyocytes (iCells)

Rat neonatal ventricular myocytes were isolated 1–2 days after birth as previously described (*Dennis et al., 2011*). Briefly, hearts were minced in HBSS, and tissue fragments were digested overnight with trypsin at 4°C. Trypsinized fragments were treated repeatedly for short periods of time with collagenase at 37°C followed by trituration. Dissociated cells were pre-plated for 2 hr at 37°C in DMEM supplemented with 5% fetal bovine serum (FBS) and penicillin/streptomycin. NRVMs were collected and replated in DMEM/5% FBS/penicillin/streptomycin with 0.1 mM bromodeoxyuridine (BrdU) to suppress fibroblast growth and maintained at 37°C, 5% CO2. These conditions were maintained for 24–36 hr, after which culture conditions deviated based on application of cells.

Human-induced pluripotent stem cell (hiPSC)-derived cardiomyocytes (iCell Cardiomyocytes; Cellular Dynamics International, Madison, WI) were cultured in iCell Cardiomyocytes Maintenance Medium (Cellular Dynamics International) in an atmosphere of 93% humidified air and 7% CO2 at 37°C. For electrophysiological recordings, 20000–40000 cardiomyocytes were plated on glass coverslips coated with 0.1% gelatin as described (*Ma et al., 2011*).

### Rat ventricular myocyte isolation

Single ventricular myocytes were isolated from adult rat hearts. Briefly, rats were anesthetized by injection of ketamin. Hearts were quickly removed and perfused via the aorta with a physiological salt solution (PSS) containing (in mmol/L) NaCl 140, KCl 5.4, MgCl2 2.5, CaCl2 1.5, glucose 11, and HEPES 5.5 (pH 7.4). After 5 min, perfusate was switched to a nominally calcium-free PSS with collagenase (Roche, 0.5 mg/mL) being added after an additional 5 min. After 15–20 min of digestion, hearts were perfused with a high K+ solution containing (in mmol/L) potassium glutamate 110, KH2PO4 10, KCl 25, MgSO4 2, taurine 20, creatine 5, EGTA 0.5, glucose 20, and HEPES 5 (pH 7.4).

Ventricles were minced in high K+ solution, and single myocytes were obtained by filtering through a 115 µm nylon mesh. Myocytes were then plated on laminin coated coverslips for 1.5 hr before fixing with 4% formaldehyde in PBS to be used for immunohistochemistry. Alternatively, cells were resuspended in a 1% formaldehyde/PBS solution to be used for ChIP studies.

## Transfection for KChIP2 overexpression, siRNA treatment, or miRNA-precursor and inhibitor delivery

NRVM cultures used for transfection and total RNA and protein collection were conducted on 35 mm dishes seeded with $1.5 \times 10^6$ cells. Following the initial 24–36 hr of plating, NRVMs were transfected with KChIP2.3 (NM_173192.2), KChIP2.4 (NM_173193.2), or KChIP2.6 (NM_173195.2) for the overexpression of KChIP2, which was inserted into the pIRES2-EGFP plasmid from Clontech as previously conducted (Deschênes et al., 2002). The plasmid without the KChIP2 insert was used as the control. Lipofectamine 2000 reagent (Invitrogen) was used to deliver the constructs according to the manufacturer's instructions. Following the transfection period, media was changed to DMEM/5% FBS/penicillin/streptomycin. Cells were cultured for 72 hr total before collection for total RNA, with a media change once after 48 hr of culture.

Knockdown of KChIP2 was conducted by transfecting with siRNA for KChIP2 (Ambion, Cat#: 4390771, ID: s132782), or a scrambled siRNA control (Ambion, Cat#: 4390843). 180 pmol of siRNA was transfected using 15 µL of Lipofectamine 2000 reagent according to the manufacturer's instructions. Following the transfection period, media was changed to DMEM/5% FBS/penicillin/streptomycin. Cells were cultured for 72 hr total before collection for total RNA, with a media change once after 48 hr of culture.

NRVM were also transfected with 180 pmol of miR-34b/c precursors (miR-34b MC12558, miR-34c MC11039, Invitrogen) or a non-targeting control (negative control 4464058, Invitrogen) using 15 µl lipofectamine RNAi Max (Invitrogen) according to the manufacturer's instructions. Cells were left for 48–72 hr and then collected for RNA. NRVM were also used for patch clamp recordings to measure $I_{Na}$ and $I_{to}$. These were plated at 100,000 cells/dish in 35 mm dishes and the miR-precursors were modified with an attached FAM reporter to visualize transfected cells. 25 pmol of miR-34 precursor with 2 µl Lipofectamine RNAiMax was used according to the manufacturer's instructions.

Transfection of control or miR-34b/c antimirs were also used during the phenylephrine induction assays for evaluation with patch-clamp recordings in NRVM and iCells and optical mapping in NRVM only. NRVM seeded at 100,000 cells/35 mm dish for patch-clamping received 22.5 pmol of miR-34b inhibitor (Invitrogen, MH12558) with 22.5 pmol of miR-34c inhibitor (Invitrogen, MH11039) or 45 pmol of a non-targeting miR-inhibitor (Invitrogen, 4464076) using 3.75 µl Lipofectamine RNAi Max (Invitrogen). NRVM used for optical mapping were seeded at $1.5 \times 10^6$ cells/35 mm dish and 225 pmol each of the miR-34b and −34c inhibitor or 450 pmol of the non-targeting control miR-inhibitor were delivered using 22.5 µl of Lipofectamine RNAi Max.

miR-inhibitors were also delivered in iCells, for cells seeded at 20,000–40,000 cells per each well of a 12-well for patch-clamp studies. 5 pmol each of the miR-34b and −34c inhibitor or 10 pmol of the non-targeting control miR-inhibitor were delivered using 1.0 µl of Lipofectamine RNAi Max.

## RNA preparation, quantitative RT-PCR of miRNA/mRNA targets, miRNA array, and whole-transcriptome microarray

Total RNA was isolated from NRVM using Trizol Reagent (Invitrogen) according to the manufacturer's instructions. RNA was also collected from human control and heart failure tissue samples. Tissue samples were first pulverized using liquid $N_2$ and mortar and pestle to assist in the homogenization with Trizol. Subsequent RNA was used as a template for cDNA synthesis in reverse transcriptase reactions using the Multiscribe Reverse Transcriptase kit (Invitrogen) for detecting both mRNAs and miRNAs. The quantitative PCR reactions were performed with the ABI 7500 Real-Time PCR system using either SYBR green technology for coding genes or Taqman reagent for detecting mature miRNAs. mRNAs were normalized with GAPDH or ribosomal protein 27 (RPL27) and miRNAs with small nucleolar RNA U87 or U6. All miRNA primer sets were designed and provided by Invitrogen Taqman Assays. Real-time PCR reactions were conducted using TaqMan Universal Master Mix II (Invitrogen). miRNA primer sets for real-time PCR detection were as follows:

Rat miR-34b: Assay name, mmu-miR-34b-5p; Assay ID, 002617; Catalogue #, 4427975

Human/Rat miR-34c: Assay name, hsa-miR-34c; Assay ID, 000428; Catalogue #, 4427975

Rat U87 (housekeeping gene): Assay name, U87; Assay ID, 001712; Catalogue #, 4427975

Human miR-34b: Assay name, hsa-miR-34b; Assay ID, 000427; Catalogue #, 4427975

Human U6 (housekeeping gene): Assay name, U6 snRNA; Assay ID, 001973; Catalogue #, 4427975

Primer sets used in the detection of mRNA transcripts were designed in Primer 3 Plus and specificity to the intended target verified using Primer Blast (NCBI).

Rat Scn5a

Forward primer, 5'-TCAATGACCCAGCCAATTACCT-3', Reverse primer, 5'-CCCGGCATCAGAGCTGTT-3'

Rat Scn1b

Forward primer, 5'-ACGTGCTCATTGTGGTGTTAACC-3', Reverse primer, 5'-CCGTGGCAGCAG-CAATC-3'

Rat Kcnd3

Forward primer, 5'-GCCTTCGAGAACCCACA-3', Reverse primer, 5'-GATCACCGAGACCGCAATG-3'

Rat Kcnip2

Forward primer, 5'-ACTTTGTGGCTGGTTTGTCG-3', Reverse primer, 5'-TGATACAGCCGTCCTTGTTGAG-3'

Rat GAPDH

Forward primer, 5'-AGTTCAACGGCACAGTCAAG-3', Reverse primer, 5'-ACTCCACGACATACTCAGCAC-3'

Rat Rpl27

Forward primer, 5'-GCTGTCGAAATGGGCAAGTT-3', Reverse primer, 5'-GTCGGAGGTGCCATCATCAA-3'

Human Kcnip2

Forward primer, 5'-TGTACCGGGGCTTCAAGAAC-3', Reverse primer, 5'-GGCATTGAAGAGAAAAGTGGCA-3'

Human Scn5a

Forward primer, 5'- CTGCGCCACTACTACTTCACCAACA-3', Reverse primer, 5'- TCATGAGGG-CAAAGAGCAGCGT-3'

Human Scn1b

Forward primer, 5'- GACCAACGCTGAGACCTTCA-3', Reverse primer, 5'- TCCAGCTGCAACACCTCATT-3'

Human Kcnd3

Forward primer, 5'- TCAGCACGATCCACATCCAG-3', Reverse primer, 5'- CTCAGTCCGTCGTCTGCTTT-3

Human GAPDH

Forward primer, 5'- TCCTCTGACTTCAACAGCGA-3', Reverse primer, 5'- GGGTCTTACTCCTTGGAGGC-3'.

RNA collected from NRVM following KChIP2 and control siRNA treatment were submitted to miRNA microarray analysis to determine miRNAs regulated by the loss of KChIP2. The array was performed by the Gene Expression and Genotyping Facility at Case Western Reserve Univesity using the Affymetrix GeneChip miRNA 4.0 array. The resulting. CEL files were used with ExpressionConsole to conduct RMA analysis to derive the relative intensities of the miRNA probe set. The raw datasets are available from the Gene Expression Omnibus (Accession GSE75806).

Additionally, RNA collected from NRVM following KChIP2 silencing with an adeno-shRNA expression system with non-targeting (control) and KChIP2 targeting constructs were used to assess global gene changes following KChIP2 loss. A total of $1.5 \times 10^6$ cells were plated on 35 mm dishes. Cells were cultured in DMEM/5% FBS/penicillin/sptreptomycin with 0.1 mM BrdU for 24 hrs. After 24 hrs, media was replaced with fresh DMEM/5% FBS/penicillin/sptreptomycin and the corresponding control and KChIP2 shRNA virus. Cells were cultured for 48 hrs (with a media change after 24 hrs) and collected for total RNA and evaluated using a whole-transcriptome microarray. The array was performed by the Gene Expression and Genotyping Facility at Case Western Reserve University using the Affymetrix rat Clariom S Assay. The resulting. CEL files were used with Expression Console and the Transcriptome Analysis Console provided by Affymetrix (available here: http://www.affymetrix.

com/support/technical/software_downloads.affx) to derive the relative changes in gene expression. The raw datasets are available from the Gene Expression Omnibus (Accession GSE94623)

## Design of shRNA viral construct

The design of non-targeting (control) and KChIP2 shRNAs was conducted as described (*Campeau et al., 2009*) with modifications. shRNA inserts were optimally designed using a compilation of the RNAi Consortium and Invitrogen design algorithms. To begin the design, oligos were ordered that contained the shRNA sequence for control: 5'- GTTGACAGTGAGCGATCTCGC TTGGGCGAGAGTAAGTAGTGAAGCCACAGATGTACTTACTCTCGCCCAAGCGAGAGTGCCTAC TGCCTC-3' and KChIP2: 5'- GTTGACAGTGAGCGCGAGCTGGGCTTTCAACTTATATAGTGAAGC CACAGATGTATATAAGTTGAAAGCCCAGCTCATGCCTACTGCCTC-3'. These oligos were modified in a PCR reaction to add on cloning sites for insertion into the pSM2 vector using the primer set: 5'- CAGAAGG<u>CTCGAG</u>AAGGTATATGCTGTTGACAGTGAGCG-3'  and  5'- CTAAAGTAGCCCC TT<u>GAATTC</u>CGAGGCAGTAGGCA-3'. Underlined are the XhoI and EcoRI restriction sites used for cloning. Following the insertion of this sequence into the pSM2 vector, the insertion was cloned out again using the primer set 5'- GAGCTC<u>GCTAGC</u>GCTACCGGTCGCCACCATGGTGAGCAAGGGC-GAGG-3' and 5'-GATTGCC<u>AAGCTT</u>CTAGATAAACGCATTAGTCTTCCAATTG-3'. Underlined are the NheI and HindII restriction sites, which were then used to clone the inserts into the adenovirus construct, Ad.CGI. During insertion, the GFP encoded by the viral vector was digested out as the shRNA system expression GFP in tandem with the shRNA. The modified Ad.GFP construct was transfected with the psi5 vector into CRE8 cells for the production and amplification of packaged viral constructs to then be used for silencing studies.

## Adult rat heart tissue fractionation to assess nuclear localization of KChIP2

Fractionation of adult rat heart tissue was performed as described (*Baghirova et al., 2015*) with slight modifications. Briefly, freshly isolated heart tissue was minced in ice cold PBS. Tissue was washed several times to remove residual blood from sample. Approximately 300 mg of tissue was weighed out and suspended in cytosolic lysis buffer, consisting of 150 mM NaCl, 50 mM HEPES (pH 7.4), 25 μg/mL Digitonin, and 10% Glycerol. Tissue pieces were homogenized then filtered through a QIAshredder homogenizer column (Qiagen, 79656). Filtered lysate was then incubated at 4°C on an end-over-end rotator for 10 min. Samples were then centrifuged at 4000 x g for 10 min at 4°C. Supernatant was collected as the cytosolic fraction. The remaining pellet was resuspended in membrane lysis buffer consisting of 150 mM NaCl, 50 mM HEPES (pH 7.4), 1% IGEPAL, and 10% glycerol. Sample was incubated for 30 min in end-over-end rotator at 4°C, followed by centrifugation at 6000 x g for 10 min at 4°C. The supernatant was collected as the membrane associated fraction, while the remaining cell pellet was resuspended in the nuclear lysis buffer consisting of 150 mM NaCl, 50 mM HEPES (pH 7.4), 0.5% sodium deoxycholate, 0.1% sodium dodecyl sulfate, and 10% glycerol. Lysate was placed on an end-over-end rotator for 10 min at 4°C, which was then followed by brief sonication. The lysate was then centrifuged at 6800 x g for 10 min at 4°C. The supernatant was collected as the nuclear fraction. Roche protease inhibitor tablets were added fresh before the addition of each lysis buffer.

## Immunoblotting

In order to perform western blot experiments looking at KChIP2 nuclear expression, cytosolic, membrane, and nuclear extracts were isolated as described above. 20–30 μg of protein extracts were loaded into SDS-PAGE gels, transferred to nitrocellulose membranes, and western blotting performed using lactate dehydrogenase (Abcam Cat# ab52488 RRID:AB_2134961, 1:1000) to represent the cytosolic fraction, Lamin-B1 (Abcam Cat# ab16048 RRID:AB_443298, 1:1000) representing the nuclear fraction, Serca2a (1:1000, Dr. Periasamy, Ohio State University) and KChIP2 (UC Davis/NIH NeuroMab Facility Cat# 75–004 RRID:AB_2280942, 1:50) to observe localization.

Western blot performed on NRVM was conducted to assess Kv4.3 protein expression following miR-34 precursor treatment. NRVM were rinsed with PBS then scraped and collected. Cell pellets were re-suspended in RIPA Buffer (150 mM sodium chloride, 1.0% NP-40 or Triton X-100, 0.5% sodium deoxycholate, 0.1% SDS (sodium dodecyl sulphate), 50 mM Tris, pH 8.0, plus Roche Inhibitor

tablet) and then sonicated on ice to disrupt cell membranes. 30–40 µg of whole cell extract was loaded into SDS-PAGE gels, transferred to nitrocellulose membrane, and western blotting performed using Kv4.3 (UC Davis/NIH NeuroMab Facility Cat# 75–017 RRID:AB_2131966, 1:500), and actin (Sigma-Aldrich Cat# A4700 RRID:AB_476730, 1:1000).

## Immunofluorescence

Freshly isolated adult rat ventricular myocytes were plated on laminin coated coverslips for 1.5 hr to allow for attachment. Cells were quickly rinsed with room temperature PBS before being fixed by 4% formaldehyde in PBS for 15 min. Cells were permeabilized for 10 min in PBS + 0.03% Triton X-100 and blocked for 2 hr in a solution of PBS, 5% normal goat serum, and 1% BSA. Cells were incubated overnight with primary antibody lactate dehydrogenase (Abcam Cat# ab52488 RRID:AB_ 2134961, 1:100) and KChIP2 (UC Davis/NIH NeuroMab Facility Cat# 75–004 RRID:AB_2280942, 1:50) in PBS with 2% normal goat serum and 1% BSA. Cells were rinsed 3x in PBS then incubated with secondary antibody (Alexa-568 Thermo Fisher Scientific Cat# A11036 RRID:AB_10563566 1:500 against LDH and Alexa-647 Thermo Fisher Scientific Cat# A-21236 RRID:AB_2535805 1:500 against KChIP2) in PBS with 2% normal goat serum and 1% BSA for 2 hr at room temperature. Coverslips were mounted onto glass slides with mounting media containing DAPI. Labeled cardiomyocytes were scanned with a Leica DMi8 confocal microscope.

## Reporter assays and designing reporter constructs

### miR-34b/c promoter reporter assays

miR-34b/c promoter reporter assays were performed in COS-7 cells (ATCC Cat# CRL-1651, RRID: CVCL_0224). $0.4 \times 10^5$ cells were plated in 24-well plates. 24 hr later cells were transfected with Polyfect transfection reagent (Qiagen) according to manufacturer's instructions. 75 ng of either the pGL4.10 promoter-less control, the pGL4.10+miR-34b/c promoter, or the pGL4.10 + 500 bp DRE deleted construct was transfected with 225 ng of KChIP2.3, KChIP2.4, KChIP2.6, or GFP control vectors. 5 ng of pGL4.74 renillin construct was co-transfected as a normalizing control. Cells were cultured for an additional 48 hr before they were prepared for measuring luciferase activity normalized to renillin.

### 3'-UTR reporter assays

3'-UTR reporter assays were performed in HEK293 cells (ATCC Cat# CRL-1573, RRID:CVCL_0045). $0.4 \times 10^5$ cells were plated in 24-well plates. 24 hr later cells were transfected with Lipofectamine 2000 (Invitrogen) according to the manufacturer's instructions. Cells were transfected with 50 ng of the pmirGlo construct containing either the 3'-UTR for *SCN5A*, *SCN1B*, or *KCND3*. Reporter constructs were co-expressed with the rno-miR-34b and −34c precursors together (0.75 pmol each precursor) or the non-targeting control precursor (1.5 pmol). The cells were cultured for an additional 48 hr, after which they were prepared for measurement of luciferase and renillin activity for normalization.

All reporter assays made use of the Promega Dual-Luciferase Reporter Assay system (Promega). Reagents and cells were prepared according to the manufacturer's instructions. Data was collected using a Perkin-Elmer EnSpire 96-well plate reader.

### Designing miR-34b/c promoter

The miR-34b/c promoter was cloned from chromatin isolated from rat liver. Forward (5'-GAGCTCGCTAGCTAAACGTGTTCACATTTTGTTGCC- 3') and reverse (5'-TGCCAAGCTTCAG TCCCCGGAGACCCTC-3') primers containing NheI and HindIII restriction sites respectively, indicated by underlined regions, were used to amplify the promoter region and allow cloning into the Promega pGL4.10 promoterless luciferase vector. Deletion of the putative DRE site identified by MatInspector software (RRID:SCR_008036) (*Cartharius et al., 2005*) was conducted using the Quick-Change II Site-Directed Mutagenesis kit (Stratagene) according to the manufacturer's instructions using the primers listed: 5'- TTAACGGAGACGGGACCCGGCGTGAG-3' and 5'- CTCACGCCGGG TCCCGTCTCCGTTAA-3'. All plasmids were sequenced to confirm the presence and integrity of inserted elements. MatInspector is a commercially available software from the Genomatix software suite (https://www.genomatix.de/).

## Designing 3'-UTR constructs

3'-UTRs for *SCN5A*, *SCN1B,* and *KCND3* genes were also cloned from rat liver chromatin. A series of primers were used to clone genomic fragments that flanked the region of the 3'-UTR containing the miR-34b/c target site into the pmirGLO Dual-Luciferase miRNA target expression vector (Promega). Scoring and identification of the target sites was done using TargetScan 7.1 (RRID:SCR_ 010845) (*Lewis et al., 2005*) (available here: http://www.targetscan.org/vert_71/). Forward and reverse primers for *SCN5A*, with XhoI and XbaI sites underlined, were: 5'- GCTAGC<u>CTCGAG</u>GCA-GAGTTCCGCGTCTCTGT-3' and 5'-GGGGCAGCTC<u>TCTAGA</u>GCTTTTAATTCTGGC-3'. Forward and reverse primers for *SCN1B*, with NheI and XbaI sites underlined, were: 5'- CTC<u>GCTAGC</u>TTCCCA-CACGCACTGCCA-3' and 5'- GAG<u>TCTAGA</u>GAGATGAGGCCCAGAACCC-3'. Forward and reverse primers for *KCND3* with the NheI and XbaI sites underlined, were: 5'-CTC<u>GCTAGC</u>GTGAGGTCACC TTAGCCGG-3' and 5'- GAG<u>TCTAGA</u>CCAGGCACAAGTCTGCAGTA-3'. Mutagenesis was conducted on the identified miR-34b/c seed region to disrupt miRNA interaction. The following primers were used with the QuickChange II Site-Directed Mutagenesis kit. *SCN5A*: 5'-AACATCTTTTTTCCA TGAACATCAGCAGTTCAGAGTCGGTCTCCTTAACCCTGAGC-3', 5'-GCTCAGGGTTAAGGAGACC-GACTCTGAACTGCTGATGTTCATGGAAAAAAGATGTT-3'; *SCN1B*: 5'-GCTTCCCACACGC TCGGGCAGGCCAGCCGGC-3', 5'-GCCGGCTGGCCTGCCCGAGCGTGTGGGAAGC-3'; *KCND3* site 1: 5'-ACCTTAGCCGGGCCCTGAGTCGGCAGCTGACCTGCACAG-3', 5'-CTGTGCAGGTCAGC TGCCGACTCAGGGCCCGGCTAAGGT-3'; *KCND3* site 2: 5'-GGACAGTAAATCCTTCTCCGTGAG TCGGAAGTACTGCAGACTTGTGCCT-3', 5'-AGGCACAAGTCTGCAGTACTTCCGACTCACGGA-GAAGGATTTACTGTCC-3'.

All plasmids were sequenced to confirm the presence and integrity of inserted elements.

## Chromatin immunoprecipitation

Chromatin Immunoprecipitation was performed as described with minor modifications (*Schmidt et al., 2009*). Briefly, freshly isolated adult rat cardiomyocytes were fixed in a 1% formaldehyde solution in PBS for 14 min and quenched with 0.125 M glycine for 5 min. Cells were treated with a 0.05% trypsin/0.02% EDTA 1x PBS solution for 8 min at 37°C to partially digest the cells aiding in removal of cytoplasmic extract and purification of nuclear extract during cell lysis steps. Trypsin was inactivated by the addition of 10% FBS, and the cell pellet was rinsed 3x in ice cold PBS. Chromatin was extracted by the treatment with several lysis buffers. Lysis buffer 1 (50 mM Hepes-KOH, Ph7.5; 140 mM NaCl; 1 mM EDTA; 10% Glyerol; 0.5% Igepal; 0.25% Triton-X) was added to the cells for 10 min with rocking, followed by 15–20 dounces with a glass teflon douncer on ice. This cell lysate fraction was discarded and the remaining cell pellet was resuspended in Lysis buffer 2 (10 mM Tris-HCl, pH 8,0, 200 mM NaCl; 1 mM EDTA; 0.5 mM EGTA) for 5 min with rocking. This was again followed by 15–20 dounces with a glass teflon douncer on ice. Lastly, remaining cell pellet was resuspended in Lysis buffer 3 (10 mM Tris-HCl, pH 8.0; 100 mM NaCl; 1 mM EDTA; 0.5 mM EGTA; 0.1% Na-Deoxycholate; 0.5% *N*-lauroylsarcosine). Cell suspension was split in half to be used for IgG or KChIP2 ChIP. Samples were then sheared on a BioRuptor (Diagenode, total 18 cycles, hi-power, 30 s on/off). The sonicated chromatin was immunoprecipitated with 15 ug of antibody (either α-KChIP2 or IgG control) bound to Dynabeads (Invitrogen) followed by washing and elution. Immuoprecipitate and input chromatin samples were then reverse crosslinked followed by purification of genomic DNA. Target and nontarget regions of genomic DNA were amplified by qRT-PCR using SYBR Green. Data were analyzed by calculating the immunoprecipitated DNA enrichment normalized to a region 8 kb upstream of the target site in the KChIP2-IP compared to the IgG-IP. Antibodies used in ChIP were KChIP2 (UC Davis NeuroMab 75–004) and IgG (Millipore Cat# 12–371 RRID:AB_145840) ChIP-PCR primer sequences were: miR-34b target site: forward 5'- GGTCACTCGGCCAGTAGGA-3', reverse 5'- GGAGTCCTGCTCTCCCTCA-3'. miR-34b 8 kb upstream: 5'- CCACCCTCTCAGTAGC TTGC-3', reverse 5'- CAGTGCCAGGGGATAGGAAG-3'

## Phenylephrine stimulation of myocytes with adenovirus or antimir treatment

Phenylephrine stimulation experiments were performed to evaluate gene expression changes, functional changes in ionic current by patch-clamp technique, or conduction properties by optical mapping. RNA studies for gene expression changes were conducted in 6-well plates with $1.5 \times 10^6$ cells

plated per well for the collection of RNA. For patch-clamp recordings, NRVMs were on coverslips coated with laminin (Sigma, L2020) inside of 35 mm dish at a density of 100,000 cells/well. For optical mapping 1.5 × 10^6 NRVMs were plated on aclar coverslips (Electron Microscopy Sciences) coated with fibronectin (BD Biosciences, 356008) in a 35 mm dish. Following the initial 24–36 hr of plating, media was switched to 1:1 DMEM:F12 (without serum or BrdU) and supplemented with 1x insulin-transferrin-selenium-X (Invitrogen), 1% PS, and 142 µM Na$^+$ Ascorbate for an additional 24–36 hr. After this time, treatment media was applied, consisting of the same DMEM:F12 media with supplements and 100 µM phenylephrine. At the same time, control cells without phenylephrine were transduced with adeno.GFP, while phenylephrine treated cells received either the adeno.GFP or adeno.KChIP2.6 to restore KChIP2 expression during phenylephrine treatment. Alternatively, cells were transfected using Lipofectamine RNAi Max using manufacturer's protocol to deliver a control or combination of miR-34b and −34c antimir. In the case of transfected cells, the transfection was performed prior to the initiation of phenylephrine treatment. Phenylephrine treatment was sustained for 48 hr (fresh media was swapped after 24 hr, maintaining phenylephrine treatment, but no more virus was applied). Phenylephrine studies were also performed on iCells. iCell Cardiomyocyte Maintenance Medium was supplemented with 142 µM Na$^+$ Ascorbate. iCells were only treated with the antimirs and submitted to patch-clamp recordings with the same treatment conditions applied to the NRVM. Notably, cells used for patch clamp recordings or optical mapping were washed at least three times over a minimum of 20 min in media without phenylephrine present for washout.

## Patch clamp experiments

Macroscopic $I_{Na}$ and $I_{to}$ were recorded using the whole-cell configuration of the patch clamp technique. $I_{Na}$ was recorded in the solution containing 50 mM NaCl (for NRVM) and 25 mM (for iCells), 80 or 105 mM N-methyl D-glucamine, 5.4 mM CsCl, 1.8 mM MgCl2, 1.8 mM CaCl2, 10 mM glucose, 10 mM HEPES, pH 7.3. 1 µM of nisodipine was used to block L-type Ca currents. $I_{Na}$ was elicited from a holding potential of −80 mV with depolarizing voltage pulses from −60 mV to 45 mV for 16 ms. To measure $I_{to}$, cells were placed in the Tyrode's solution containing (mmol/L) NaCl 137, KCl 5.4, CaCl$_2$ 2.0, MgSO$_4$ 1.0, Glucose 10, HEPES 10, CdCl$_2$ 0.3, and TTX 100 mM, pH to 7.35 with NaOH. Patch pipettes were pulled from borosilicate capillary glass and lightly fire-polished to resistance 0.9–1.5 MΩ when filled with electrode solution composed of (mmol/L) aspartic acid 120, KCl 20, MgCl$_2$ 2, and HEPES 5, NaCl 10, EGTA 5, Na-GTP 0.3, Phosphocreatine 14, K-ATP 4, Creatine phosphokinase two and brought to a pH of 7.3. $I_{to,total}$ amplitude was measured as the difference between peak current and steady-state current during a 400 ms voltage step ranging from –30 to +60 mV from a holding potential of –70 mV. Recording $I_{to,f}$ used a modified protocol to kinetically isolate the current. A 150 ms voltage step to −80 mV from a holding potential of −20 mV was used to allow recovery of $I_{to,f}$ but not $I_{to,s}$. This was followed by a 50 ms prepulse to −20 mV to eliminate $I_{Na}$. $I_{to,f}$ amplitude was then measured as the difference between peak current and steady-state current during 500 ms voltage steps ranging from −30 to +40 mV. Ionic current density (pA/pF) was calculated from the ratio of current amplitude to cell capacitance. All experiments were performed at 35°C except $I_{Na}$ (room temperature). Low-resistance electrodes (<2 MΩ) were used, and a routine series resistance compensation was performed to values of >80% to minimize voltage clamp errors. The uncompensated Rseries was therefore <2 MΩ. Command and data acquisition were operated with an Axopatch 200B patch clamp amplifier controlled by a personal computer using a Digidata 1200 acquisition board driven by pCLAMP 7.0 software (Axon Instruments, Foster City, CA). Current densities, cell capacitance, current-voltage relationship, and conductance, were measured as previously described (*Shinlapawittayatorn et al., 2011*).

## Optical mapping studies

Following 48 hr of PE treatment of the NRVM, cells were prepared for optical mapping studies. Prior to recordings, NRVMs were washed twice for 10 min each in DMEM:F12 treatment media without PE to wash out the PE and remove any acute effects. They were then transferred to Tyrodes solution (140 NaCl, 4.56 KCl, 0.73 MgCl$_2$, 10 HEPES, 5.0 dextrose, 1.25 CaCl$_2$) containing 10 µM Di4 (Sigma, D8064) for 20 min. Monolayers were then washed with normal Tyrodes solution before mounting on stage adapter to maintain cells at 34–35°C. Di4 fluorescence 685/80 nm was measured using an upright microscope (MVX10, Olympus) with a cooled CCD camera (Princeton Instruments). A solid-

state light source (Sola Light Engine, Lumencore) was used for dye excitation (510/80 nm) over a 16 × 12 mm field of view. Cells were paced by point stimulation at cycle lengths of 1000 ms, 750 ms, 500 ms, 350 ms and 350 ms to obtain conduction velocity and APD restitution curves. Analysis of recordings were conducted via custom software developed in Matlab (MathWorks) as described previously (PMID: 12960954). Additional Matlab custom software (Rhythm) was also used for analysis (*Laughner et al., 2012*). Arrhythmia data was collected using baseline pacing (S1, 750 ms) followed by a single premature stimulus (S2) with a coupling interval beginning at 150 ms and prolonged by 10 ms until either capture of a single beat or arrhythmia ensued.

## Ethics statement and tissue acquisition

This study was carried out in strict accordance with the recommendations in the Guide for the Care and Use of Laboratory Animals of the National Institutes of Health. The protocol for tissue isolation from neonatal rat (Protocol Number: 2013–0015) was approved by the Committee on the Ethics of Animal Experiments of Case Western Reserve University. Tissue from the left ventricular free wall of non-failing and failing human heart samples were acquired from the Cleveland Clinic Foundation (CCF) tissue repository. All protocols were approved by the CCF Institutional Review Board (IRB# 2378). Samples were received coded and no identifying metrics were documented for the study.

## Statistical testing

Results are expressed as mean ± SEM and represent data from at least three independent experiments. Statistical analysis for continuous data was performed using a two-tailed Student's t-test. When multiple comparisons were evaluated, a Bonferroni correction was performed. The null hypothesis was rejected if $p < 0.05$. Statistical testing of non-continuous data, as seen with arrhythmia susceptibility measurements, was performed using the Mann-Whitney Test. Evaluation of samples sizes were initially performed using stringent conditions for expected molecular and functional changes. Assuming as a little as a 20% change in control to treated conditions, an error rate of 10%, and a power of 0.8 at a threshold of 0.05, provided a sample size of 4 per experimental condition. However, because of anticipated variability from the use of primary cells for many of the experiments and multiple comparisons in some datasets, larger sample sizes were used.

## Acknowledgements

This work was supported by NIH R01 (R01HL096962 and R01HL132520) (ID), an American Heart Association Pre-Doctoral Fellowship from the Great Rivers Affiliate 13PRE17060106 (DMN), and a T32 Training Grant HL105338-01 (DMN).

## Additional information

### Funding

| Funder | Grant reference number | Author |
|---|---|---|
| National Institutes of Health | R01HL096962 | Isabelle Deschênes |
| National Institutes of Health | HL105338-01 | Drew M Nassal |
| American Heart Association | 13PRE17060106 | Drew M Nassal |
| National Institutes of Health | R01HL132520 | Isabelle Deschênes |

The funders had no role in study design, data collection and interpretation, or the decision to submit the work for publication.

### Author contributions

DMN, Conceptualization, Data curation, Formal analysis, Funding acquisition, Validation, Investigation, Visualization, Methodology, Writing—original draft, Writing—review and editing; XW, Data curation, Formal analysis, Investigation, Methodology, Writing—review and editing; HL, Investigation; DM, Data curation, Formal analysis, Investigation, Methodology; AR-N, Investigation, Methodology; CSM, Resources, Supervision, Writing—review and editing; EF, Conceptualization,

Supervision, Methodology, Project administration; KRL, Conceptualization, Resources, Formal analysis, Supervision, Methodology, Project administration, Writing—review and editing; ID, Conceptualization, Resources, Supervision, Funding acquisition, Validation, Investigation, Methodology, Writing—original draft, Project administration, Writing—review and editing

### Author ORCIDs
Drew M Nassal, http://orcid.org/0000-0002-8018-3913
Isabelle Deschênes, http://orcid.org/0000-0002-1812-7267

### Ethics

Human subjects: Tissue from the left ventricular free wall of non-failing and failing human heart samples were acquired from the Cleveland Clinic Foundation (CCF) tissue repository. All protocols were approved by the CCF Institutional Review Board (IRB# 2378). Samples were received coded and no identifying metrics were documented for the study. As such, no consent to publish was obtained by the patients for this manuscript.

Animal experimentation: This study was carried out in strict accordance with the recommendations in the Guide for the Care and Use of Laboratory Animals of the National Institutes of Health. The protocol for tissue isolation from neonatal rat (Protocol Number: 2013-0015) was approved by the Committee on the Ethics of Animal Experiments of Case Western Reserve University, with every effort being made to minimize suffering of animals during the isolation procedure.

## Additional files

### Supplementary files
• Supplementary file 1. List of genes regulated from microarray following KChIP2 silencing.

### Major datasets
The following datasets were generated:

| Author(s) | Year | Dataset title | Dataset URL | Database, license, and accessibility information |
|---|---|---|---|---|
| Deschenes I | 2015 | miRNA expression data from neonatal rat ventricular myocytes (NRVM) silenced for KChIP2 | https://www.ncbi.nlm.nih.gov/geo/query/acc.cgi?acc=GSE75806 | Publicly available at NCBI Gene Expression Omnibus (accession no: GSE75806) |
| Deschenes I | 2017 | mRNA expression data from neonatal rat ventricular myocytes (NRVM) silenced for KChIP2 | https://www.ncbi.nlm.nih.gov/geo/query/acc.cgi?acc=GSE94623 | Publicly available at NCBI Gene Expression Omnibus (accession no: GSE94623) |

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
