## [Decision Letter]

Thank you for submitting your article "KChIP2 is a Core Transcriptional Regulator of Cardiac Excitability" for consideration by *eLife*. Your article has been reviewed by three peer reviewers, one of whom is a member of our Board of Reviewing Editors, and the evaluation has been overseen by Harry Dietz as the Senior Editor. The reviewers have opted to remain anonymous.

The reviewers have discussed the reviews with one another and the Reviewing Editor has drafted this decision to help you prepare a revised submission.

Summary:

The manuscript by Nassal and Deschenes and colleagues addresses the mechanisms of arrhythmogenesis in patients with heart disease. They describe a transcriptional repression role for the proteins KChIP2 – previously known to interact at the membrane with and regulate the Kv4.3 potassium channel in a Ca-dependent manner, and identifies genes for mir34b and c as targets of this transcriptional repression. These miRs regulate channel subunits transcripts responsible for *I_Na_* and *I_to_ (SCN5*A, *SCN1B* and *KCND3* itself) through degradation and translational repression mechanisms. Thus, this work proposes a new arm of a regulatory network in the heart controlled by the multifunctional proteins KChIP2, which is known to be significantly repressed in heart disease. The idea that KChIP2 can be a transcriptional regulator is derivative of similar findings on a highly related channel-regulator KChIP3/DREAM active in neuronal cells, which has been characterized in detail. Nonetheless, KChIP2 expression is specific to the heart, giving this study specific context, and the elucidation of the down stream miR pathway is novel. The data appear for the most part robust and modulation of miR23bc and channel subunits appears to be confirmed in human failing heart samples, although here controls are missing. The second part of the paper addresses the functional implications of this pathway in an in vitro model of cardiac disease (PE treated NRVMs). They provide a surrogate that appear to support the model of critical involvement of the above pathway in dampening of *I_Na_* and *I_to_* currents in disease. Overexpression of miR34b and c precursors dampens currents (*I_to_* only in human cells as there is compensatory expression of Kv4.2 in mouse CMs), while over expression of KChIP2 restores currents dampened after PE treatment of NRVC. A strong finding is that point pacing induced arrhythmogenic activity in monolayer cultures treated with PE and this can be blocked by antimeres to miR34b/c.

While the concept of KChIP2 as a transcriptional repressor is interesting and important for our understanding of the stressed and failing heart, the evidence to support the findings is incomplete.

Essential revisions:

1) Figure 1 The representation of the miRNA profiling in Figure 1 is misleading: fold change on the y axis pronounces upregulated miRNAs, but hides downregulated miRNAs; please use log2 fc vs detection or a scatter plot. Downregulation of several miRNAs upon KChIP2 knock-down is less compatible with KChIP2 as a transcriptional repressor, thus it should be clearly visible in the figure. A list of all significantly regulated miRNAs more than two-fold up/down would also be appreciated by the community. In addition analysis of miRNA is fine; however it is unlikely that a transcription factor specifically regulates only these miRNAs. If KChIP2 is a transcription factor, a more thorough analysis of transcriptional changes upon acute/chronic down- or upregulation of KChIP2 would be potentially interesting and support the findings.

2) Cell fractionation indicates presence of some KChIP2 in the nuclear fraction (Figure 1). Cell fractionation experiments using overexpressed proteins can be misleading. Membrane bound/membrane associated proteins should be used as a control here to exclude the possibility of contamination of the nuclear fraction. In addition the experiment would be far more convincing if native protein would be detected in this assay – is this possible? The images included in Figure 1—figure supplement 1 are not completely convincing – Z-scans of the confocal images would be appreciated; again immunofluorescence of native KChIP2 would be superior.

3) Data for human cells are not complete. The authors should include more comprehensive data for the human system: (i) is the DRE and its position in the promoter of miR-34 conserved in humans? (ii) are the miR-34b/c binding sites conserved in the UTRs of human orthologues of *SCN5B/SCN1B/KCND3*? (iii) In samples of failing human heart reduction of KChIP2 and elevation of miR-34b/c was observed, but surprisingly, no data on human Sodium or Potassium channels were included into the manuscript (Figure 4). In addition it remains unclear whether the observed changes are due to changes in transcription or due to changes in cellular composition in the failing heart. Carefully considered additional controls should be included to exclude such artifacts, for example other channel genes not targeted by miR34.

4) Antimirs were applied to NRVM in culture and excitability in monolayers was measured. It would be interesting to see the result of PE stimulation and subsequent antimir-treatment in vivo. See Bernardo 2012 PNAS. Such experiments could also be used to demonstrate modulation of the KChIP2-miR-34b/c-ion channel axis in vivo.

5) The paper would benefit from more extensive study of KChIP2 DNA-binding. ChIP PCR in transfected cells is prone to artifacts; ChIP-seq would look much more convincing and again native protein would be preferred in case of ChIP experiments.

6) Most disappointing is that the functional outcome from altering miR-34 is very minor. In particular, the changes in *I_to_* in NRVM are statistically not significant, as noted likely because Kv4.2 is the dominant channel in rat CMs. However, the issues is: despite restoring *I_Na_* by miR-34 antimir in NRVMs, conduction velocity (CV) was not restored. Thus it is not clear how reentrant arrhythmia can be prevented under miR-34 anitimir treatment.

7) Although the authors claim that miR-34 mostly affects Kv4.3 and not KV4.2, the same group reported earlier that KChIP2 altered both Kv4.2 and Kv4.3, which implies that there could be an additional pathway by which KChIP2 can influence cardiac repolarization. Although I agree that the present study provides strong evidence supporting KChIP2 as a transcriptional repressor through modulating miR-34, the discussion may need to state limitations of this study and potential alternative regulation of *I_to_* factors other than miR-34.

---

## [Author Response]

Essential revisions:

1) Figure 1 The representation of the miRNA profiling in Figure 1 is misleading: fold change on the y axis pronounces upregulated miRNAs, but hides downregulated miRNAs; please use log2 fc vs detection or a scatter plot. Downregulation of several miRNAs upon KChIP2 knock-down is less compatible with KChIP2 as a transcriptional repressor, thus it should be clearly visible in the figure.

This is a notable point as there are an appreciable number of miRNAs that are also downregulated in response to KChIP2 siRNA treatment. While for the purposes of our investigation we were interested in the miRNAs experiencing upregulation from the loss of transcriptional repression provided by KChIP2, there are clearly miRNAs that are also down- regulated and should be portrayed equally. For the miRNAs where a down-regulation is observed, it is unlikely that these miRNAs would be direct targets of KChIP2 transcriptional activity, as they would not be consistent with its mode of transcriptional repression. They are most likely indirect targets from a modified state within the cardiomyocytes, such as altered Ca^2+^ handling because of changes to the cardiac action potential through the loss in KChIP2. None- the-less, it is valuable to identify what targets are being influenced and Figure 1 has been modified to show those changes.

A list of all significantly regulated miRNAs more than two-fold up/down would also be appreciated by the community.

These tables have been added as Figure 1 to the manuscript to provide this data.

In addition analysis of miRNA is fine; however it is unlikely that a transcription factor specifically regulates only these miRNAs. If KChIP2 is a transcription factor, a more thorough analysis of transcriptional changes upon acute/chronic down- or upregulation of KChIP2 would be potentially interesting and support the findings.

In the pursuit of more fully characterizing the transcriptional state of KChIP2 in the heart, we agree that it would be interesting to understand what other transcriptional targets might be directly influenced by KChIP2. Importantly though, the primary focus of this manuscript was the identification of the KChIP2/miR-34b/c axis of regulation and its implications in electrical remodeling of the heart. Given the degree of verification and confirmation in identifying KChIP2 transcriptional targets, particularly considering its multimodal function, a full gene analysis following KChIP2 loss presents with the challenging task of understanding which are direct targets of transcriptional activity and which are secondary responses. However, gene changes following KChIP2 loss either by direct or indirect means are still substantial in understanding molecular changes that might be precipitated by KChIP2 loss during heart failure remodeling. Therefore, to begin populating a list of genes that are altered in response to acute KChIP2 loss, a whole-transcriptome microarray was performed on neonatal rat cardiomyocytes treated with an shRNA to acutely suppress KChIP2 expression. A list of significantly up- and down-regulated mRNAs has been provided as a table in Figure 1—figure supplement 1C and D. Importantly, we saw the conservation of decreases in both *SCN5A* and *SCN1B*. At the same time, we see that other highly predicted miR-34b/c targets such as Sema4b are also reduced. In total, 293 genes experienced at least a 2 fold increase in expression and 407 genes a 2 fold decrease. Notably, of the genes experiencing increased expression, 192 of them (65.5%) were predicted to contain a DRE motif within promoter elements, implicating the potential of KChIP2 transcriptional activity directly mediating these changes. This is consistent with the role of KChIP2 as a transcriptional repressor and highlights its potential involvement in transcriptionally influencing a large portion of these gene changes.

Together, this provides a shortened list of genes which might be influenced by KChIP2 transcriptional activity, as well as genes which may be targeted by the pathologic KChIP2 loss occurring in the numerous conditions of heart disease. Indeed, numerous gene pathways relevant to cardiac disease remodeling were implicated by the changes in these gene targets, suggesting that KChIP2 loss could be a precipitating event in disease progression. Pursuing these targets more extensively will be the scope of further studies. Importantly though, the gene array results further support and reinforce our findings that KChIP2 acts as a transcription factor and we thank the reviewers for this suggestion.

*2) Cell fractionation indicates presence of some KChIP2 in the nuclear fraction (Figure 1). Cell fractionation experiments using overexpressed proteins can be misleading. Membrane bound/membrane associated proteins should be used as a control here to exclude the possibility of contamination of the nuclear fraction. In addition the experiment would be far more convincing if native protein would be detected in this assay – is this possible? The images included in Figure 1*—*Figure 1 are not completely convincing – Z-scans of the confocal images would be appreciated; again immunofluorescence of native KChIP2 would be superior.*

We appreciate the concern of the reviewers over the methods used to identify the nuclear expression of KChIP2. Previous manuscripts have shown through immunohistochemistry the presence of KChIP2 staining in the nucleus of native cardiomyocytes, and so our primary incentive originally was to identify if different KChIP2 isoforms aligned with a distinguishable pattern of nuclear expression. However, heterologous over-expression of a protein in non-native cells could absolutely change the normal behavior and trafficking of proteins. In order to address these concerns, cell fractionation experiments were replaced with tissue from native rat hearts as suggested by the reviewers. We now show in Figure 1 the presence of KChIP2 in fractions representing cytoplasmic, membrane, and nuclear lysates. Moreover, we used Serca2a immunoblotting as a representative protein for potentially contaminating membrane fractions.

Importantly, Serca2a was absent from the nuclear fraction, in addition to lactate dehydrogenase (LDH) representing the cytosolic fraction, showing that KChIP2 nuclear expression is not the consequence of contaminating fractions.

Further, immunohistochemistry experiments were updated to include staining in native adult rat cardiomyocytes (Figure 1). LDH was again used as a cytosolic marker to ensure the absence of cytosolic leak into nuclear regions. The use of z-stacks shows clearly the omission of LDH from the nucleus, as marked by DAPI, while also revealing enrichment of KChIP2 in the nuclear regions relative to cytosolic levels. Together these results suggest KChIP2 maintains nuclear expression in a native state, consistent with its role as a transcriptional repressor.

3) Data for human cells are not complete. The authors should include more comprehensive data for the human system: (i) is the DRE and its position in the promoter of miR-34 conserved in humans? (ii) are the miR-34b/c binding sites conserved in the UTRs of human orthologues of SCN5B/SCN1B/KCND3?

Given the inclusion of human derived cardiomyocytes in our current set of figures, we appreciate the reviewers’ suggestion that we include comparisons of the genetic elements responsible for the functional data that we show. When comparing the conservation of the DRE site between rat and human, we do see that the DRE site is maintained upstream of miR-34b in the human promoter (Figure 5). Specifically, in the rat, the DRE motif resides 254 bp upstream of the

miR-34b stem-loop, while in human it is 242 bp upstream, suggesting that loss of KChIP2 in human heart failure tissue is also responsible for the elevation in miR-34b/c that is observed.

As for the conservation of the miR-34b/c target sites, comparisons between rat and human show the conservation of the seed region for *SCN5A* and *KCND3*, but no conservation for *SCN1B* (Figure 5). The loss of the seed region in the human background is reflected in the human heart failure data for gene expression, showing a minimal and non-significant loss in *SCN1B* expression (Figure 5). Never-the-less, our functional data in human derived cardiomyocytes show restoration of *I_Na_* following miR-34b/c block through targeting of *SCN5A*.

(iii) In samples of failing human heart reduction of KChIP2 and elevation of miR-34b/c was observed, but surprisingly, no data on human Sodium or Potassium channels were included into the manuscript (Figure 4). In addition it remains unclear whether the observed changes are due to changes in transcription or due to changes in cellular composition in the failing heart. Carefully considered additional controls should be included to exclude such artifacts, for example other channel genes not targeted by miR34.

In order to reinforce the relationship between miR-34b/c regulation of ion channel remodeling in human, we added data on gene transcript changes in normal and failing human heart tissue. We see significant loss in expression of *SCN5A* and *KCND3* in failing heart samples. Figure 5 has been updated to show these changes. The loss of these transcripts and the increase in miR-34b/c parallels the remodeling we see in the rat with KChIP2 loss directly or through phenylephrine stimulation, reflecting the relevance of this pathway in human disease.

This significance is reflected in our figures showing blockade of miR-34b/c in human derived cardiomyocytes leads to rescue of both *I_Na_* and *I_to_*. Additionally, we added recordings of the rapid delayed rectifier current, *I_Kr_* (Figure 6—figure supplement 1). In cardiac disease remodeling, this current frequently experiences a loss in expression, leading to compromised repolarization and APD prolongation. We show that in response to PE stimulation in the human derived cardiomyocytes, this current experienced, as expected, significant reductions. However, with the treatment of miR-34b/c antimirs, *I_Kr_* was not rescued, reinforcing the specificity of miR-34b/c regulation, as opposed to altering the overall molecular state of the cell. Moreover, this specificity is also illustrated by the varying scenarios of *I_to_* rescue between NRVM and human derived cells. Rat *I_to_* is encoded by the three channel genes Kv1.4, Kv4.2, and Kv4.3, whereas miR-34b/c is predicted to target only Kv4.3. Therefore, the rescue of only Kv4.3 following miR- 34b/c block, did not lead to a significant rescue of *I_to,total_*, but did resolve a partial rescue of *I_to,f_*. However, in iCells®, Kv4.3 is predominant, and we were able to observe a full rescue of the current even when assessing *I_to,total_*. Addition of these new results confirms the specificity of the transcriptional regulation.

4) Antimirs were applied to NRVM in culture and excitability in monolayers was measured. It would be interesting to see the result of PE stimulation and subsequent antimir-treatment in vivo. See Bernardo 2012 PNAS. Such experiments could also be used to demonstrate modulation of the KChIP2-miR-34b/c-ion channel axis in vivo.

We agree with the reviewers that being able to observe these mechanisms in an in vivo state of regulation would be interesting. After beginning these experiments, however, we have realized that this is an extensive endeavor and a significant amount of time is required to have the powered experiments necessary for evaluation of in vivo model and as well as proper statistical analysis. Most challenging is that simply getting animals into a state of heart failure that develops the appropriate phenotypic changes and pathological electrical remodeling necessary to evaluate our mechanisms and therapeutic interventions requires nearly 5 months time. We also must identify therapeutic ranges and proper dosing of miR-34 blockade for evaluating our time courses and outcomes, which means that having a consistent treatment protocol further delays the onset of the 5 month protocol. Additionally, due to the breadth of the work required and the information that will be obtained, we believe that these additional extensive studies will be easily a stand-alone manuscript. As a result we regrettably will not be able to provide in vivo analysis within the scope of this manuscript.

However, we believe the impact of the mechanisms investigated within the current manuscript are not diminished by the absence of in vivo data. We set out to understand if KChIP2 could act as a transcriptional repressor and how acute KChIP2 loss could lead to the changes in gene expression of *SCN5A, SCN1B*, and *KCND3*, particularly given that KChIP2 loss and the currents encoded by these ion channel subunits are diminished in multiple states of cardiac pathology. To that end, the current data has provided a clear and robust relationship of this axis of regulation, which at the same time has identified a novel transcriptional action of KChIP2 extremely relevant for mediating disease signaling and the development of established arrhythmic substrates. Moreover, we continue to see the observed relationship between these genes in human heart failure tissue, suggesting that they are relevant changes in an in vivo state of diseased myocardium.

5) The paper would benefit from more extensive study of KChIP2 DNA-binding. ChIP PCR in transfected cells is prone to artifacts; ChIP-seq would look much more convincing and again native protein would be preferred in case of ChIP experiments.

We appreciate the reviewers concern over the identification of KChIP2 DNA binding through the use of cardiomyocytes over-expressing KChIP2. Initially, we had settled on this approach due to expected limitations in the experimental design which could not be easily addressed. Most ChIP studies are critically dependent on the availability of an appropriate ChIP-grade antibody for maximizing the efficiency of pulldown. However, as we are characterizing novel KChIP2 DNA binding, there is no available antibody designed for this purpose. At the same time, NRVM were chosen over adult myocytes due to the easier disruption and isolation of nuclear content, without cytosolic contamination, assisting in a more efficient sonication of chromatin material, as well as being able to shed non-DNA bound KChIP2 residing in the cytosol. However, NRVM express native KChIP2 at a dramatically lower amount, therefore, we had chosen to over-express the protein. Regardless, artificial expression of KChIP2 does have the potential to create artifacts, and so to satisfy that concern we optimized our ChIP-PCR workflow to handle native adult rat cardiomyocytes. The results of those efforts revealed a significant enrichment of the miR-34b/c promoter following KChIP2 pull-down compared to IgG control in native adult rat myocytes without needing to overexpress KChIP2 (new Figure 2), reinforcing that this transcriptional interaction is indeed a native event.

We agree that the most effective strategy at identifying genome wide contributions to KChIP2 transcriptional activity would be to perform ChIP-seq. We had invested significant time and resources into performing ChIP-seq studies prior to the submission of this manuscript to understand the full scope of KChIP2 binding. However, we consistently ran into difficulties at the point of library generation. Consultations with ChIP-seq experts suggested that without a proper ChIP grade antibody or alternatively a tagged fusion protein, we would not achieve sufficient chromatin enrichment for successful library generation. While a tagged fusion was a possible alternative, we would still run into the issue of having to over-express the construct, while at the same time run the risk of modifying the native behavior of the cell by addition of the tag. Lastly, a transgenic animal with the fusion could be made to maintain endogenous gene levels, however, that would be beyond the scope of this manuscript at this time. Due to each of these issues, we therefore chose to proceed with ChIP-PCR in native adult rat myocytes, where discrete DNA regions can be evaluated for enrichment with lower starting quantities of DNA.

Additionally, the primary motivation behind this current study was whether KChIP2 transcriptional activity was involved in the regulation of *SCN5A, SCN1B*, and *KCND3*. To that end, with the addition of evaluating native KChIP2 protein, we have successfully shown this event does occur. Moreover, this pathway has shown its importance in electrical remodeling relevant to the development of arrhythmogenesis, highlighting not just the novel role for KChIP2, but one with a significant contribution to disease remodeling.

6) Most disappointing is that the functional outcome from altering miR-34 is very minor. In particular, the changes in I_to_ in NRVM are statistically not significant, as noted likely because Kv4.2 is the dominant channel in rat CMs. However, the issues is: despite restoring I_Na_ by miR-34 antimir in NRVMs, conduction velocity (CV) was not restored. Thus it is not clear how reentrant arrhythmia can be prevented under miR-34 anitimir treatment.

We appreciate the reviewers’ comments regarding the overall significance of miR-34 antimir treatment and its influence in arrhythmia susceptibility. As noted, because of the use of rat cardiomyocytes and the targeting of only Kv4.3, we were only able to minimally restore *I_to_*, leading to no changes in APD for disease remodeled cardiomyocytes (PE treated cells). While we were able to target *I_Na_*, our expectation was also that we would be able to restore conduction velocity. However, the fact that we did not suggests additional remodeling to the cardiomyocytes that is not influenced by miR-34b/c, which may include cellular coupling. Notably, from our gene array, Cx43 mRNA expression was reduced by 67% following KChIP2 silencing, though it is not a target of miR-34b/c. However, the observation of a trend toward increased conduction velocity, as well as the significant rescue of the relative refractory interval following miR-34b/c blockade, represents preservation of sodium channel availability, which very dramatically had an effect on arrhythmia inducibility.

While APD and conduction velocity are major substrates in arrhythmogenesis, we believe that conduction velocity is not the primary mechanism responsible for eliciting the reentrant events observed. In this instance of arrhythmia susceptibility, the development of the reentrant loop occurred at the point of excitation as a result of unidirectional conduction block of the premature stimulus due to heterogenous excitability. As sodium channel excitability becomes compromised, there becomes an expanding window of vulnerability where a premature stimulus at the right time will capture and propagate in one direction, but will still be in a state of refractoriness in another, establishing the substrates necessary for reentry to occur. Both mathematical and experimental data are in support of this mechanism (1-4). With enough sodium channel in reserve, this vulnerability window is minimal and the site will either experience complete block or capture. By restoring *I_Na_* we are therefore reducing the probability of reentry as a direct consequence of the premature stimulus.

This mechanism of conduction block leading to reentry is mechanistically distinct from one created as a result of conduction slowing, but still extremely relevant to the overall arrhythmic risk. Physiologically, an ectopic beat from an event such as a delayed after depolarization could lead to a premature stimulus that if occurring within this vulnerable window would establish the same events leading to the formation of reentry. Therefore, we believe that the observations made after miR-34b/c blockade are, in fact, major contributors to arrhythmic risk as observed in Figure 7, and consistent with the mechanisms identified throughout the manuscript.

7) Although the authors claim that miR-34 mostly affects Kv4.3 and not KV4.2, the same group reported earlier that KChIP2 altered both Kv4.2 and Kv4.3, which implies that there could be an additional pathway by which KChIP2 can influence cardiac repolarization. Although I agree that the present study provides strong evidence supporting KChIP2 as a transcriptional repressor through modulating miR-34, the discussion may need to state limitations of this study and potential alternative regulation of I_to_ factors other than miR-34.

The reviewers are correct in this observation. There are multiple modes of regulation by which KChIP2 can target the family of Kv4 channels and we appreciate the opportunity to clarify these mechanisms. Numerous publications have investigated the physical interaction KChIP2 has with both Kv4.2 and Kv4.3, leading to the enhanced trafficking and increased stabilization of the channels (30). With KChIP2 loss, the consequence has been reduced expression of both Kv4 channels. Therefore, while we highlight a transcriptional capacity of KChIP2 to target miR-34b/c and indirectly target Kv4.3, KChIP2 is also destabilizing both Kv4.3 in addition to Kv4.2 through its physical interaction. These distinctions are highlighted well in the experiments where we treat the NRVM with phenylephrine and either restore KChIP2 with adenovirus or instead treat with miR-34b/c antimirs. We see that with miR-34b/c block, we are unable to effectively restore *I_to_* in rodent cells, but are successful in doing so in human derived cells where Kv4.3 is more dominant. However, when we restore KChIP2 in the NRVM, we are able to restore *I_to_* because KChIP2 can directly stabilize Kv4.2 expression as well. Potential limitations of these distinctions will be especially important when considering the species of animal being investigated and whether or not KChIP2 expression or miR-34b/c activity is being targeted.

However, the observation that KChIP2 can influence the same target through multiple pathways only potentiates its importance in disease remodeling. This material is also included in the Discussion.